# UBQLN2 restrains the domesticated retrotransposon PEG10 to maintain neuronal health in ALS

Holly H Black[1†], Jessica L Hanson[2†], Julia E Roberts[1], Shannon N Leslie[1], Will Campodonico[1], Christopher C Ebmeier[1], G Aaron Holling[1], Jian Wei Tay[3], Autumn M Matthews[1], Elizabeth Ung[1], Cristina I Lau[1], Alexandra M Whiteley[1]*

[1]Department of Biochemistry, University of Colorado Boulder, Boulder, United States; [2]Institute for Behavioral Genetics, University of Colorado Boulder, Boulder, United States; [3]Biofrontiers Institute, University of Colorado Boulder, Boulder, United States

**Abstract** Amyotrophic Lateral Sclerosis (ALS) is a fatal neurodegenerative disease characterized by progressive motor neuron dysfunction and loss. A portion of ALS cases are caused by mutation of the proteasome shuttle factor *Ubiquilin 2* (*UBQLN2*), but the molecular pathway leading from UBQLN2 dysfunction to disease remains unclear. Here, we demonstrate that UBQLN2 regulates the domesticated gag-pol retrotransposon 'paternally expressed gene 10 (PEG10)' in human cells and tissues. In cells, the PEG10 gag-pol protein cleaves itself in a mechanism reminiscent of retrotransposon self-processing to generate a liberated 'nucleocapsid' fragment, which uniquely localizes to the nucleus and changes the expression of genes involved in axon remodeling. In spinal cord tissue from ALS patients, PEG10 gag-pol is elevated compared to healthy controls. These findings implicate the retrotransposon-like activity of PEG10 as a contributing mechanism in ALS through the regulation of gene expression, and restraint of PEG10 as a primary function of UBQLN2.

## Editor's evaluation

In this manuscript, the investigators provide evidence that levels of Paternally Expressed Gene 10 (PEG10) protein are regulated by Ubqln2 and that proteolytic fragments from PEG10 cleavage induce changes in gene expression, in particular genes that encode proteins involved in axon biology. These data along with their finding that PEG10 levels are increased and alterations of proteins regulated by PEG10 are found in the spinal cord of ALS patients support a role for the abnormal induction of PEG10-regulated genes in ALS.

*For correspondence:
alexandra.whiteley@colorado.edu

†These authors contributed equally to this work

## Introduction

ALS is a fatal neurodegenerative disease that typically presents in mid-life and is characterized by a progressive loss of motor function (*Brown and Al-Chalabi, 2017*). In many cases, loss of motor function is accompanied by frontotemporal dementia (FTD), characterized by cognitive impairment, behavioral changes, and loss of executive function (*Couratier et al., 2017*). Of all ALS cases, 90% are sporadic (sALS), while the remaining 10% are familial (fALS) and can be traced to mutations in a variety of genes, including the proteasome shuttle factor *UBQLN2* (*Deng et al., 2011*; *Gorrie et al., 2014*; *Williams et al., 2012*). Animal models of fALS have led to discoveries that are broadly applicable to both fALS and sALS, including the involvement of oxidative stress, RNA binding proteins, and protein aggregation in ALS-mediated neuronal dysfunction (*Ferraiuolo et al., 2011*; *Peters et al., 2015*); however, the specific molecular pathways that lead to disease remain poorly understood.

In a previous global proteomic study, animal models of *UBQLN2*-mediated fALS revealed a dramatic accumulation of the domesticated retrotransposon PEG10 in diseased tissue (*Whiteley et al., 2021*). Domesticated retrotransposons are a class of genes that encode virus-like proteins which have lost the ability to replicate and have evolved adaptive functions (*Volff, 2006*). *PEG10*, which is necessary for placental development (*Ono et al., 2006*), is one of a family of domesticated retrotransposon genes derived from the Sushi-ichi lineage of Ty3/Gypsy LTR retrotransposons and codes for both *gag* and *pol* domains separated by a programmed ribosome frameshifting site (*Brandt et al., 2005*). Use of the ribosomal frameshift occurs with high efficiency, resulting in two forms of PEG10 proteins: gag, and gag-pol (*Clark et al., 2007*; *Lux et al., 2010*; *Manktelow et al., 2005*). The overwhelming share of research on PEG10 has focused on its contributions to placental development (*Abed et al., 2019*; *Ono et al., 2006*) and cancer progression (*Akamatsu et al., 2015*; *Kim et al., 2019*); however, PEG10 has also recently been implicated in the neurological disease Angelman's syndrome (*Pandya et al., 2021*). Here, we describe a novel potential role for PEG10 in the neurodegenerative disease ALS due to its ability to accumulate in the absence of functional UBQLN2, leading to changes in the expression of neuronal genes.

## Results

### UBQLN2 exclusively regulates degradation of the frameshifted gag-pol PEG10

*UBQLN2* is one of five human Ubiquilin (*UBQLN*) genes that facilitate proteasomal degradation of 'client' proteins (*Hjerpe et al., 2016*; *Itakura et al., 2016*; *Lee and Brown, 2012*; *Suzuki and Kawahara, 2016*; *Whiteley et al., 2017*; *Zheng et al., 2020*) via an N-terminal protein domain which binds to the proteasome (*Finley, 2009*; *Saeki, 2017*), and a C-terminal domain which binds to ubiquitin (*Zhang et al., 2008*). All five *UBQLNs* have similar protein domain architecture and amino acid sequences, and are widely assumed to have shared client populations. The most notable difference in Ubiquilins is their tissue expression profile: *UBQLN1* and *UBQLN4* are ubiquitously expressed, *UBQLN3* and *UBQLNL* are expressed almost exclusively in the testes, and *UBQLN2* is uniquely enriched in neural and muscle tissues (*Marín, 2014*). Additionally, *UBQLN2* is unique among Ubiquilins for containing a small, proline-rich PXX repeat region that is commonly mutated in *UBQLN2*-mediated fALS (*Deng et al., 2011*). To test the specificity of the previously observed relationship between UBQLN2 and its client PEG10, human embryonic stem cells (hESCs) lacking *UBQLN1*, *UBQLN2*, or *UBQLN4* genes (*Figure 1—figure supplement 1*) were probed by western blot for endogenous PEG10 protein expression (*Figure 1A*). *UBQLN2*^-/- hESCs were the only cell line that demonstrated an increase in PEG10 protein, indicating that PEG10 is exclusively a client of UBQLN2, and no other UBQLN. Furthermore, only the gag-pol form of PEG10 accumulated, while the gag form remained unchanged upon perturbation of any *UBQLN* gene (*Figure 1B–C*).

Next, we sought to determine whether UBQLN2 was sufficient to control PEG10 abundance. To test this, we used a HEK293 cell line lacking expression of three abundant *UBQLN* genes: *UBQLN1*, *UBQLN2*, and *UBQLN4* (referred to as triple knockout, or 'TKO' cells) (*Itakura et al., 2016*) and quantified endogenous PEG10 by western blot (*Figure 1E*). TKO cells had more than twice the amount of PEG10 gag-pol protein in cell lysate as compared to WT cells, while gag protein levels were unchanged (*Figure 1F–G*). In contrast, TKO cells expressing an inducible WT *UBQLN2* construct were able to restrain PEG10 gag-pol to the level seen in WT cells (*Figure 1F–G*). Paired with the hESC data, we conclude that UBQLN2 is necessary and sufficient for the restriction of PEG10 gag-pol levels.

Mutations in *UBQLN2* lead to fALS (*Deng et al., 2011*; *Williams et al., 2012*), and are thought to cause both a loss of degradative function (*Chang and Monteiro, 2015*; *Le et al., 2016*; *Wu et al., 2020*) as well as a toxic gain of function by promoting misfolded UBQLN2 self-assembly (*Dao et al., 2019*; *Sharkey et al., 2020*; *Sharkey et al., 2018*). To test the ability of mutant *UBQLN2* to restrain PEG10, TKO cells were also complemented with two known ALS-causing *UBQLN2* missense mutant alleles (*Deng et al., 2011*). Expression of any of the *UBQLN2* alleles reduced levels of gag-pol compared to TKO cells (*Figure 1F*), indicating that the two mutant *UBQLN2* alleles retain some level of gag-pol degradation. Consistent with a partial loss of function, mutant *UBQLN2*^P506T-expressing cells had significantly higher endogenous levels of PEG10 gag-pol than WT *UBQLN2*-rescued TKO cells (*Figure 1F*). In comparison, rescue with *UBQLN2*^P497H was indistinguishable from WT *UBQLN2*

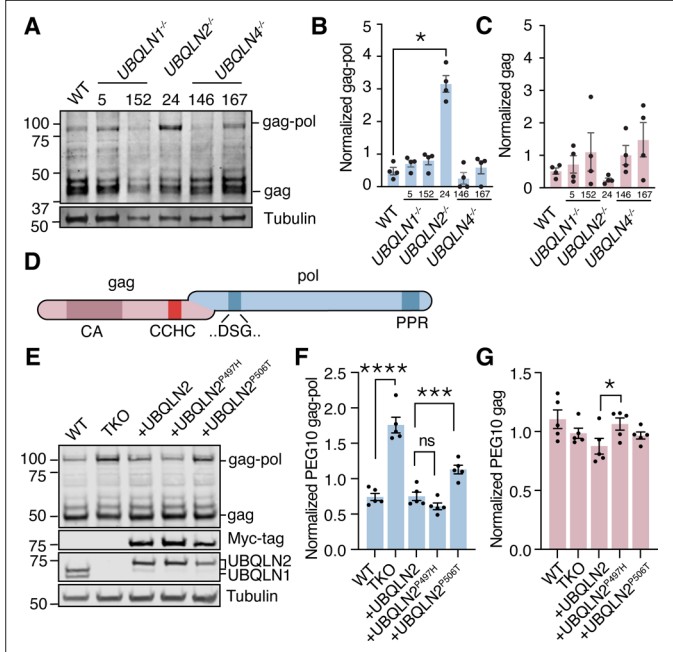

**Figure 1.** Ubiquilin 2 (UBQLN2) regulates paternally expressed gene 10 (PEG10) gag-pol abundance. (**A**) Human embryonic stem cells (ESCs) had individual Ubiquilin (*UBQLN)* genes deleted by CRISPR gene editing and clones were probed by western blot for endogenous PEG10 protein. Full-length gag-pol protein accumulates only upon *UBQLN2* loss. n=4 independent experiments. Two independently generated knockout lines are shown for UBQLN1 and UBQLN4; clone identification numbers are shown at top of the blot. (**B–C**) Quantification of gag-pol (**B**) and gag (**C**) abundance in hESC cell lines of (**A**). PEG10 protein was normalized to Tubulin, then normalized to the average intensity for each individual experiment. n=4 independent experiments and significance was determined by multiple comparisons test. Mean ± SEM is shown. No differences in gag (**C**) were detected with an ordinary one-way ANOVA. (**D**) Schematic of PEG10 protein. The first reading frame (gag) contains a capsid-like (CA) region, as well as a retroviral zinc finger ('CCHC'). The pol-like sequence contains a retroviral-type aspartic protease with one active site 'DSG' motif, as well as a C-terminal polyproline repeat domain (PPR). (**E**) WT or TKO cells stably transfected with doxycycline- (dox-) inducible constructs expressing *Myc-UBQLN2*, *Myc-UBQLN2^{P497H}*, or *Myc-UBQLN2^{P506T}* were probed for endogenous PEG10. (**F**) Quantitation of gag-pol abundance in mutant *UBQLN2*-expressing cells. Gag-pol was normalized to Tubulin and to the average intensity of each experiment. Mean ± SEM is shown for each condition. n=5 wells per condition collected from two different passages. (**G**) Quantitation of gag abundance. Shown is the mean ± SEM of n=5 wells. Multiple comparison tests were run with Bonferroni correction to compare WT and triple knockout (TKO) cells as well as WT OE with the two mutant lines. *p<0.05, **p<0.01, ***p<0.001, ****p<0.0001.

The online version of this article includes the following source data and figure supplement(s) for figure 1:

**Source data 1.** Uncropped blot for *Figure 1A*.

**Source data 2.** Uncropped blot for *Figure 1E*.

**Figure supplement 1.** Validation of human embryonic stem cells (hESC) Ubiquilin (UBQLN) loss.

**Figure supplement 1—source data 1.** Uncropped blot for *Figure 1—figure supplement 1A*.

**Figure supplement 1—source data 2.** Uncropped blot for *Figure 1—figure supplement 1B*.

---

expression in this assay (*Figure 1F*). In all cases, gag levels were not dramatically elevated by mutant *UBQLN2* expression (*Figure 1G*).

## PEG10 abundance in UBQLN2-deficient cells reflects defects in proteasome-mediated degradation

Ubiquilins regulate protein abundance by facilitating interactions between the proteasome and protein destined for breakdown (*Zheng et al., 2020*). To examine the degradation of PEG10, we performed a cycloheximide chase assay where WT and TKO cells were transfected with tagged PEG10, protein synthesis was halted with cycloheximide, and protein abundance was followed over time by western

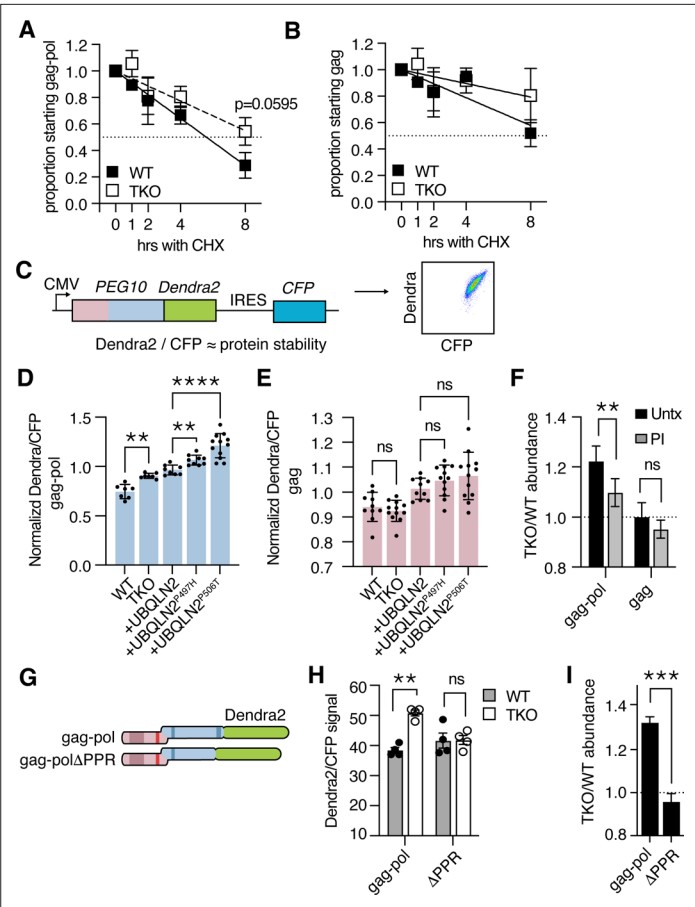

**Figure 2.** Ubiquilin 2 (UBQLN2) facilitates proteasome-dependent degradation of paternally expressed gene 10 (PEG10) gag-pol through the C-terminus of PEG10. (**A–B**) WT (filled squares) and UBQLN TKO (open squares) HEK cells were transfected with an HA-tagged form of PEG10 expressed under the control of a CMV promoter, treated with cycloheximide, and chased for 8 hr followed by western blot. (**A**) The half-life of PEG10 gag-pol is between 4–8 hr for WT cells, and more than 8 hr for UBQLN-deficient cells. (**B**) The half-life of the PEG10 gag is longer than 8 hr for WT and TKO cells. For (**A–B**) n=3 independent experiments and mean ± SEM is shown. No timepoint was deemed significantly different between WT and triple knockout (TKO) cells by paired t-test; p-value of 8 hr gag-pol timepoint (**A**) is shown. (**C**) Schematic of PEG10 protein abundance reporter. PEG10 is fused at the 3′ end to Dendra2, followed by an IRES-CFP. Right: example dot plot showing Dendra2 and CFP signal in transfected cells. (**D–E**) Dendra2 over CFP MFI ratio for PEG10 gag-pol (**D**) and gag (**E**) in WT and *UBQLN1*, *2*, and *4* 'TKO' HEK293 cells. TKO cells were rescued with the expression of WT or mutant UBQLN2 alleles as in *Figure 1E–G*. Significance was determined by multiple comparisons test and mean ± SEM is shown of n=7 independent experiments. (**F**) WT and TKO cells were transfected with either gag-pol or gag and incubated in the presence or absence of proteasome inhibitor (PI) for 12 hr. The ratio of Dendra2/CFP was determined for each cell population and the normalized ratio of PEG10 for TKO/WT cells is shown. Values over 1.0 indicate dependence on *UBQLNs* for restriction. Significance was determined by Student's t-test. n=3 independent experiments. (**G**) WT and truncation mutant of PEG10 gag-pol fused to the fluorophore Dendra2. ΔPPR is missing the last 27 amino acids containing the polyproline repeat. (**H**) Protein abundance of PEG10-Dendra2 fusions was determined for WT (filled circles) and TKO (open circles) cells by flow cytometry n=4 independent experiments. Significance was determined by multiple comparisons test and mean ± SEM is shown. (**I**) TKO/WT abundance values for data in (**H**). Values over 1.0 indicate dependence on *UBQLNs* for restriction. Mean ± SEM is shown. Significance was determined by Student's t-test. *p<0.05, **p<0.01, ***p<0.001, ****p<0.0001.

The online version of this article includes the following figure supplement(s) for figure 2:

**Figure supplement 1.** Paternally expressed gene 10 (PEG10) and Ubiquilin 2 (UBQLN2) specificity.

blot. The approximate half-life of gag-pol protein in WT cells was between 4 and 8 hr, whereas the half-life of gag-pol in TKO cells was beyond the 8 hr timepoint of our chase experiment (*Figure 2A*). In contrast, the half-life of gag protein was longer than 8 hr for both WT and TKO cell lines (*Figure 2B*). In each case, the differences in PEG10 abundance between WT and TKO cells did not reach significance (*Figure 2A–B*).

Because of the difficulty in performing kinetic assays on proteins with long half-lives, we sought an alternative readout of protein abundance that could be linked to protein degradation. In line with recently used reporters (*Itakura et al., 2016*; *Manford et al., 2021*; *Whiteley et al., 2021*), PEG10 was fused at the C-terminus to the fluorescent protein Dendra2 (*Klementieva et al., 2016*), followed by an IRES-CFP cassette, to generate a transfection-controlled measure of protein abundance (*Figure 2C*, *Figure 2—figure supplement 1A*). PEG10-reporter constructs were then transfected into WT and TKO HEK293 cells to examine the abundance of PEG10 upon *UBQLN* deficiency (*Figure 1D–E*). TKO cells accumulated significant amounts of Dendra2-tagged PEG10 gag-pol compared to WT cells (*Figure 2D*), while Dendra2-tagged PEG10 gag was unchanged (*Figure 2E*). In this model system, re-introduction of WT UBQLN2 expression was not sufficient to reduce gag-pol levels to that of WT cells (*Figure 2D*), but was significantly better than mutant UBQLN2 bearing a P497H or P506T mutation (*Figure 2D*). Neither mutation had an effect on gag protein levels (*Figure 2E*). From this, we conclude that both disease-associated mutations result in a loss of UBQLN2's ability to restrain transfected PEG10 gag-pol levels.

To determine how the proteasome contributes to UBQLN2-dependent restriction of gag-pol abundance, cells were transfected with Dendra2-tagged constructs and incubated in the presence or absence of the proteasome inhibitor, bortezomib. After 12 hr, the cells were harvested for analysis by flow cytometry. In the absence of proteasome inhibition, gag-pol protein accumulates in UBQLN TKO cells as compared to WT cells, shown by a ratio >1 of protein abundance (*Figure 2F*). Upon treatment with a proteasome inhibitor, a significant proportion of UBQLN-dependence is lost (*Figure 2F*), indicating that much of the accumulation is proteasome-dependent. In contrast, gag protein did not accumulate in TKO cells and was not significantly changed when cells were treated with a proteasome inhibitor.

## Contribution and conservation of the C-terminal polyproline region (PPR) to PEG10 restriction

As UBQLN2 selectively regulated proteasomal degradation of the gag-pol form of PEG10, we hypothesized that a unique region of the pol domain rendered it dependent on UBQLN2 for its degradation. The gag region contains a retroviral capsid domain and CCHC-type zinc finger (*Figure 1D*). The pol region of PEG10 is less well understood but contains an aspartic protease domain (*Clark et al., 2007*) and a 27 AA C-terminal PPR region containing twelve prolines in tandem, and 18 in total (*Clark et al., 2007*; *Figure 1D*). To identify a region of PEG10 necessary for UBQLN-dependent degradation, either PEG10 gag-pol, or a construct lacking the C-terminal PPR, was expressed in HEK293 cells (*Figure 2G*). While gag-pol protein was more abundant in TKO cells compared to WT cells, PEG10 lacking the PPR failed to accumulate (*Figure 2H–I*). Removal of the PPR resulted in insignificant changes to protein abundance in WT cells, and a decrease in protein abundance in TKO cells (*Figure 2H*). From this, we conclude that the PPR contributes to PEG10's ability to evade traditional proteasomal degradation pathways: when present, it renders the cell dependent on UBQLNs to facilitate degradation. When absent, the protein is capable of being degraded by pathways shared with WT cells, including traditional proteasomal degradation (*Figure 2—figure supplement 1B*).

To complement the flow-based abundance assay, cycloheximide chase experiments were also performed with the ΔPPR construct. This showed that there was still a modest increase in the half-life of PEG10 protein in TKO cells by 8 hr, though there was considerable variability in the behavior of this construct (*Figure 2—figure supplement 1C*). Together, these results identify the PEG10 PPR as a necessary region for Ubiquilin-dependent restriction.

*UBQLN2* is the most recent gene duplication event of the *UBQLN* family. The *UBQLN2* gene is only found in eutherian mammals, commonly referred to as 'placental' mammals (*Figure 3A*; *Marín, 2014*); similarly, the *PEG10* family of retrotransposon genes inserted into the mammalian genome just before the split of marsupials and eutherians (*Figure 3A*; *Brandt et al., 2005*). While marsupial and eutherian *PEG10* share homology throughout most of the gag-pol sequence, marsupials lack the

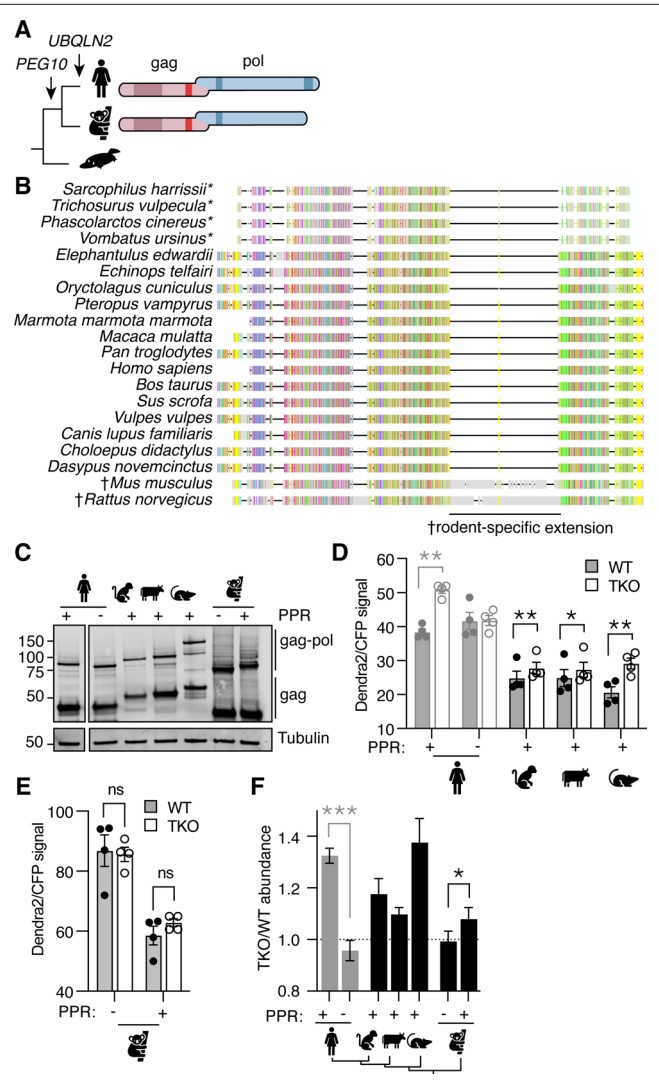

**Figure 3.** Phylogenetic investigation of the Ubiquilin 2 (UBQLN2)-paternally expressed gene 10 (PEG10) relationship. (**A**) Evolutionary schematic of PEG10 protein in eutherian and marsupial mammals. Monotremes (bottom) do not contain *PEG10* or *UBQLN2* genes. *PEG10* and *UBQLN2* appearances are highlighted with arrows. PEG10 schematic highlights the lack of C-terminal polyproline repeat region in marsupials. (**B**) Amino acid alignment of mammalian PEG10 from a diversity of mammalian species showing general conservation and lack of C-terminal polyproline domain in marsupials (starred). Colors represent the conservation of aligned amino acids; the proline is yellow. † highlights the rodent-specific extension within pol. (**C**) Western blot demonstrating expression of mammalian PEG10 gag and gag-pol in WT cells. PEG10 was detected by N-terminal HA-tag. (**D–E**) Dendra2-green over CFP MFI ratios for PEG10 from various placental mammals (**D**) and the marsupial Koala (**E**) in WT (filled circles) and triple knockout (TKO) (open circles) cells. Shown is the mean ± SEM from four independent experiments, with triplicate transfection wells for each. Human is shown in gray and is duplicated from *Figure 2* for ease of visualization. Statistics were determined by multiple comparisons tests. (**F**) The ratio of Dendra2/CFP was determined for each cell population in (**D–E**) and the normalized ratio of PEG10 for TKO/WT cells is shown. Values over 1.0 indicate dependence on *UBQLNs* for restriction. Significance was determined by unpaired (human) or paired (koala) Student's *t*-test. n=4 independent experiments. For all experiments, the mean ± SEM is shown. *p<0.05, **p<0.01, ***p<0.001, ****p<0.0001.

The online version of this article includes the following source data for figure 3:

**Source data 1.** Uncropped blot for *Figure 3C*.

PPR at the C-terminus of pol (*Figure 3A–B*). PEG10 gag-pol constructs derived from all of the tested eutherian mammals accumulated in *UBQLN*-deficient TKO cells (with a TKO/WT value >1), indicating reliance on Ubiquilins for degradation (*Figure 3C, D and F*). Koalas do not have a *UBQLN2* gene, and unlike the eutherian mammals tested, Koala PEG10 gag-pol did not depend on UBQLN expression for its regulation (*Figure 3E*). However, when the human PPR was appended to the C-terminus of Koala PEG10 gag-pol, its overall abundance decreased dramatically, indicating a role for the human PPR in regulating PEG10 abundance in a Ubiquilin-independent fashion (*Figure 3E*). When the ratio of PEG10 abundance in TKO/WT cells was compared between Koala PEG10 ± PPR, we observed a significant difference between the two, reflecting a minor contribution of UBQLNs to the difference in abundance of Koala PEG10 (*Figure 3F*). In concert with data from human PEG10, we conclude that the PPR of PEG10, which is unique to eutherian mammals, is necessary for its relationship with Ubiquilins, but is not sufficient to confer complete dependence on Ubiquilins for degradation.

## Human PEG10 gag-pol self-processes like a retrotransposon

The highly specific regulation of gag-pol by UBQLN2 led us to examine the unique biological properties of this protein in more depth. The pol region of PEG10 contains a retroviral aspartic protease domain with a classic 'DSG' active site motif (*Figure 1D*; *Clark et al., 2007*), which in the ancestral Ty3 retrotransposon results in self-cleavage of the capsid (CA) and nucleocapsid (NC) protein fragments with distinct functions (*Clemens et al., 2011*; *Kirchner and Sandmeyer, 1993*; *Larsen et al., 2008*; *Sandmeyer and Clemens, 2010*). Like Ty3, PEG10 has been reported to self-cleave (*Clark et al., 2007*; *Lux et al., 2005*), and recent reports suggested that gag was cleaved between the capsid and zinc finger regions to generate two resultant gag protein fragments (*Golda et al., 2020*). Transfection with an HA-tagged form of PEG10 showed that in addition to the expected gag and gag-pol bands, there were two HA-positive lower molecular weight bands, which we hypothesized were products of self-cleavage (*Figure 4A*). When the active site aspartate of the PEG10 protease was mutated to alanine to disrupt proteolytic activity (gag-pol[ASG]), we observed a total disappearance of the lower molecular weight HA-tagged bands (*Figure 4A*), indicating that the protein products were dependent on PEG10 protease activity. Detailed biochemical analysis and bioinformatic prediction were then performed to identify the precise sites of PEG10 self-cleavage (*Figure 4B–D*). Together, the results suggested that PEG10 cleaves itself in two locations: AA114-115, and AA260-261. The N-terminal cleavage halves the capsid region, and the C-terminal cleavage generates a zinc-finger containing fragments reminiscent of retrotransposon and retroviral nucleocapsids (*Figure 4E*). Furthermore, like its retrotransposon ancestors, PEG10 gag-pol was capable of cleaving PEG10 gag in trans (*Figure 4F*), which suggested that a large pool of proteolytic products of gag could be generated from gag-pol activity.

Traditionally, retrotransposon and retrovirus gag-pol self-cleavage is necessary to complete the viral lifecycle. For example, proteolytic liberation of the Ty3 retrotransposon nucleocapsid from gag is necessary for proper capsid or virus-like particle (VLP) assembly (*Larsen et al., 2008*; *Sandmeyer and Clemens, 2010*). The PEG10 gag protein has been shown to form VLPs (*Abed et al., 2019*; *Segel et al., 2021*) that resemble those formed by retrotransposons and the gag-like gene *Arc/Arg3.1 Ashley et al., 2018*; *Pastuzyn et al., 2018*; therefore, we hypothesized that PEG10 self-cleavage may be necessary for proper VLP formation and release. PEG10 was overexpressed in cells and VLPs were harvested from the cultured supernatant by ultracentrifugation. Abundance of VLPs was then probed by western blot. Self-cleavage was not a prerequisite for PEG10 VLP release, as gag and gag-pol[ASG] were capable of releasing VLPs with similar efficiency (*Figure 4G–H*).

To observe the liberated NC protein directly, we generated a polyclonal antibody directed against the predicted NC cleavage product, encompassing AA259-325 of PEG10 gag. Western blot of HA-tagged gag-pol, gag-pol[ASG], gag, and NC using this antibody clearly showed the presence of gag-pol and gag, as well as a number of protease-dependent bands of high molecular weight that were not previously visible in the HA blot (*Figure 4I*). Because the NC has been cleaved from CA and CA[NTD] fragments, neither of these were visible by western blot; however, a new fragment derived from gag that is independent of protease activity was visible between 20–25 kDa for all constructs examined (*Figure 4I*). HA-tagged NC was weakly visible between 10–15 kDa despite identical transfection conditions (*Figure 4I*). Upon overexpression, a band at 10 kDa, corresponding to the approximate molecular weight of an isolated NC fragment, became visible only upon transfection

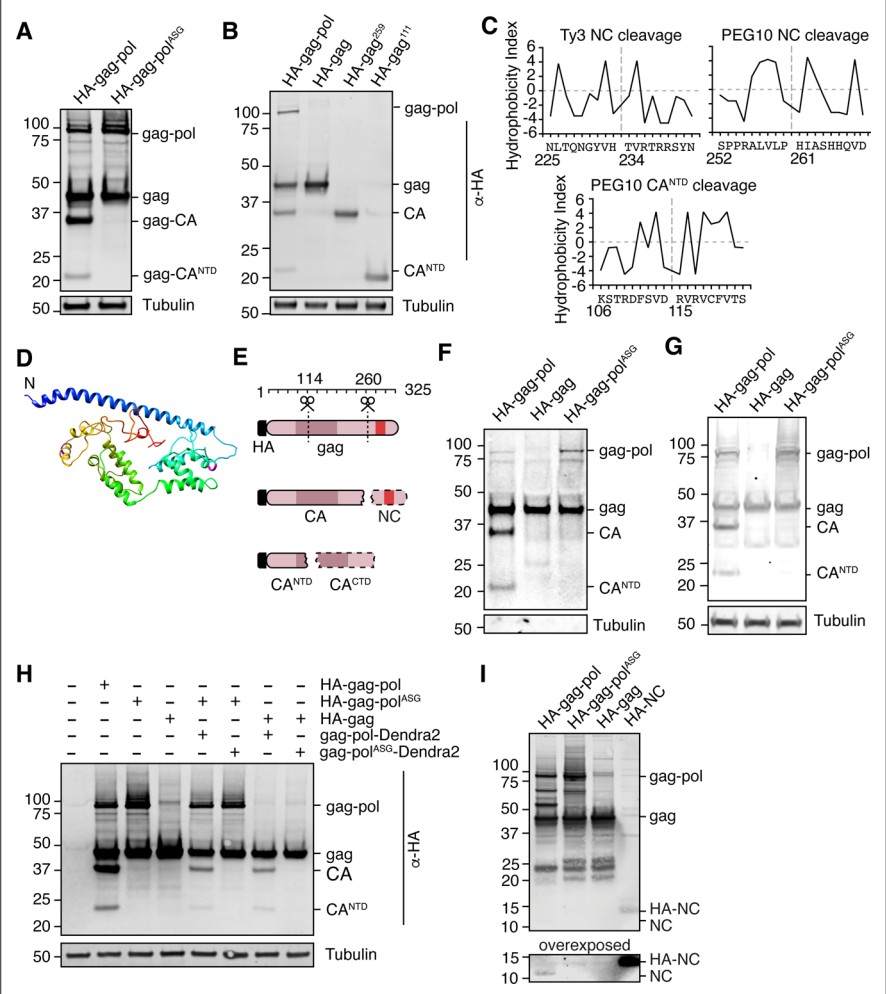

**Figure 4.** Paternally expressed gene 10 (PEG10) self-cleaves to generate a classical 'NC' fragment. (**A**) Mutation of the active site aspartic acid in the protease domain results in the disappearance of cleaved PEG10 products. N-terminally HA-tagged PEG10 was expressed either as WT or 'ASG' protease mutant in cells and probed by western blot for HA. Based on estimated molecular weight, the fragments are estimated to encompass gag-capsid (CA) and gag-capsid(NTD) (CA$^{NTD}$) fragments. n=4 independent experiments. (**B**) HA-tagged, C-terminally truncated forms of PEG10 were expressed in cells to approximate self-cleavage sites via Molecular weight. n=3 independent experiments. (**C**) Kyte-Doolittle (K-D) hydropathy analysis of Ty3 (left, similar to that in ***Kirchner and Sandmeyer, 1993***) and PEG10 nucleocapsid (right) and CA$^{NTD}$ (bottom) cleavages. Hydropathy measurements for each amino acid were plotted on a scale from P9 to P9'. Amino acid locations are shown below the amino acid alignment. Estimated cleavage locations by molecular weight in ***Figure 3D*** are close in proximity to those estimated by K-D analysis. (**D**) Structure prediction of PEG10 gag (blue N-terminus) shows that estimated cleavage sites appear accessible by protease. AA114-115 (purple) and AA260-261 (magenta) are highlighted to show estimated cleavages. (**E**) Model of PEG10 self-cleavage. PEG10 cleaves gag to generate a liberated nucleocapsid (NC) fragment. PEG10 also cleaves the gag-CA domain into CA$^{NTD}$ and CA$^{CTD}$. Dotted lines indicate that the fragments are not visible by western blot due to the absence of the N-terminal HA tag. (**F**) PEG10 gag is capable of being cleaved by PEG10 gag-pol in trans. HA-tagged and PEG10-Dendra2 fusion constructs were co-transfected into cells and the presence of HA-tagged cleavage products was assessed by western blot. n=2 independent experiments. (**G**) Presence of cleaved PEG10 products in virus-like particles (VLPs). VLPs were isolated from PEG10-transfected HEK cells by ultracentrifugation and probed for cleavage products by western blot. Tubulin was used as a control for contamination of the conditioned medium with cell fragments. n=3 independent experiments. (**H**) Western blot of accompanying cell lysate from VLP preparations. n=3 independent experiments. (**I**) Visualization of the endogenously-cleaved NC fragment by western blot. HEK cells were transfected with the listed constructs and prepared for western blot using a custom-generated antibody against PEG10 AA259-325. HA-tagged NC can

*Figure 4 continued on next page*

*Figure 4 continued*

be seen at ~14 kDa, and endogenously cleaved NC is at ~11 kDa and is only visible upon expression of cleavage-competent gag-pol.

The online version of this article includes the following source data for figure 4:

**Source data 1.** Uncropped blot for *Figure 4A*.

**Source data 2.** Uncropped blot for *Figure 4B*.

**Source data 3.** Uncropped blot for *Figure 4F*.

**Source data 4.** Uncropped blot for *Figure 4G*.

**Source data 5.** Uncropped blot for *Figure 4H*.

**Source data 6.** Uncropped blot for *Figure 4I*.

with cleavage-competent gag-pol protein (*Figure 4I*). From this, we conclude that a cleavage product consistent with an isolated NC fragment is generated upon self-cleavage by PEG10.

Proteolytic self-processing enables novel functions for domains found in the gag and gag-pol polyproteins. For the Ty3 retrotransposon, the liberation of nucleocapsid from gag regulates the localization of capsid assembly (*Larsen et al., 2008*). We hypothesized that liberated PEG10 nucleocapsid may have similarly unique localization and function following self-cleavage. To test this, individual HA-tagged PEG10 cleavage products were expressed in cells and their localization was examined by confocal microscopy. All PEG10 proteins (gag, gag-pol, CA, and nucleocapsid) were similarly expressed and localized to the cytoplasm. Intriguingly, however, only nucleocapsid was also observed in the nucleus (*Figure 5A*). To test whether the HA-tag was contributing to nuclear localization, a tagless NC construct was generated and imaged using the polyclonal NC antibody. Untransfected cells showed only a background signal with the polyclonal antibody, whereas both HA-tagged and untagged, transfected NC fragments were localized to the nucleus (*Figure 5B*). These data suggest that the self-processing of PEG10 may reveal novel functions of its proteolytic products.

## PEG10 nucleocapsid induces changes in gene expression

The nucleocapsid fragment contains a retroviral CCHC-type zinc finger that has been reported to bind DNA (*Steplewski et al., 1998*) as well as RNA (*Abed et al., 2019*; *Segel et al., 2021*). This, paired with the movement of liberated nucleocapsid to the nucleus, raised the possibility that PEG10 self-cleavage may induce unique transcriptional changes. To test this hypothesis, HEK cells were

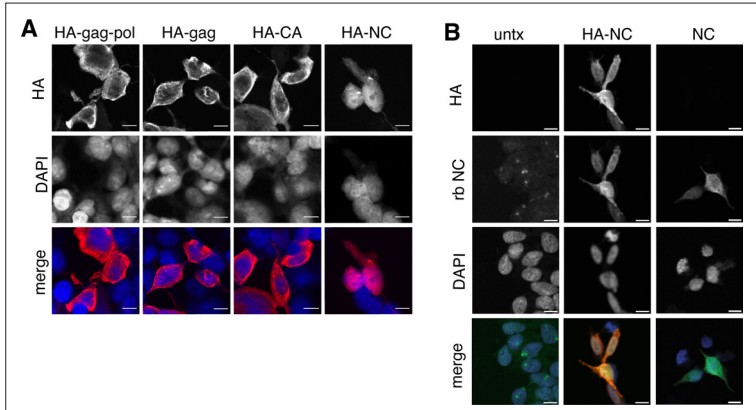

**Figure 5.** Isolated NC is uniquely found in the nucleus. (**A**) Localization of paternally expressed gene 10 (PEG10) fragments using tagged antibodies. Cells were transfected with HA-tagged PEG10 constructs, stained, and imaged by confocal microscopy. Scale bar 10 μm. HA has been colored red and DAPI blue in merged images. Shown are representative cells from 10 fields of view of each construct. n=3 independent experiments. (**B**) Localization of PEG10 NC fragment using custom NC antibody. Cells were transfected either with HA-tagged NC or untagged NC construct and stained with the listed antibodies. Scale bar 10 μm. HA has been colored red, NC green, and DAPI blue in merged images. Shown are representative cells from 10 fields of view of each construct. n=1 experiment.

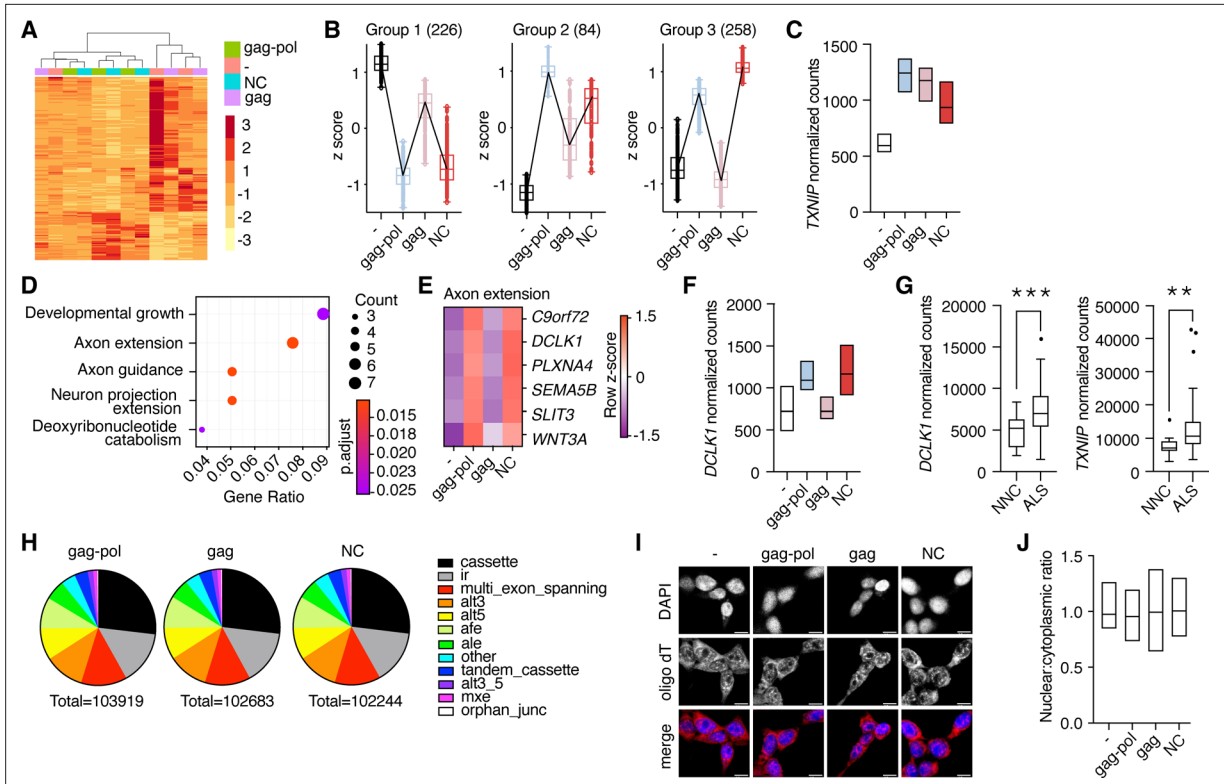

**Figure 6.** Liberated nucleocapsid alters transcription of axon extension genes. (**A**) Heatmap showing Euclidean clustering of expression profiles of HEK cells transfected with PEG10 constructs with top 200 altered genes between gag-pol and negative control plasmid. (**B**) Cluster profiling of gene expression effects as measured by RNA-seq analysis upon paternally expressed gene 10 (PEG10) construct overexpression. The number of genes in each group is listed in parentheses. Data are shown as box and whiskers min to max with the line at the median. (**C**) Normalized counts of *TXNIP* transcript from RNA-seq analysis of three biological replicates from PEG10 transfected or control transfected cells. (**D**) Top gene expression changes in NC-transfected cells by GO-term enrichment analysis. The top five GO-terms ranked by adjusted p-value are shown. Adjusted p-value is shown by color, and the size of the datapoint reflects the number of genes enriched in the pathway. (**E**) Heatmap of genes from the Axon extension GO-term showing Row z-score for each gene in the pathway. (**F**) Normalized counts of *DCLK1* from RNA-seq analysis of PEG10-transfected cells. (**G**) RNA-seq data from the Target Amyotrophic Lateral Sclerosis (ALS) dataset of post-mortem lumbar spinal cords were analyzed for *DCLK1* (left) and *TXNIP* (right) counts. NNC = non-neurological control. n=17 NNC and 127 ALS samples. For (**C,F**), data are shown as min-max floating bars with the line at mean and significance were determined by DESeq2. For (**G**), data are shown as a 5–95% box and whisker plot, and significance was determined by DESeq2. (**H**) Splice pattern changes upon PEG10 overexpression compared to control. Nucleocapsid expression does not alter patterns of splice alteration compared to gag-pol or gag expression. Splice changes were quantified and classified by MAJIQ analysis (*Vaquero-Garcia et al., 2016*). Total splice alteration counts are shown below in pie charts. ir = intron retention; afe/ale = alternative first/last exon; mxe = mutually exclusive exons. (**I**) Oligo-dT FISH showed no changes to bulk mRNA trafficking upon PEG10 overexpression. Scale bar 10 µm. Shown are representative images from 10 recorded fields of view. n=3 independent experiments. (**J**) Quantification of oligo-dT signal in the nucleus versus cytosol for each condition imaged in (**I**) showing no changes to mRNA trafficking upon PEG10 overexpression. Quantification was performed on a minimum of 60 images from each condition and is representative of 3 independent experiments. *p<0.05, **p<0.01, ***p<0.001.

The online version of this article includes the following source data and figure supplement(s) for figure 6:

**Source data 1.** RNA-seq results.

**Figure supplement 1.** Transcriptional effects of paternally expressed gene 10 (PEG10) nucleocapsid transfection resemble gag-pol, but not gag.

transfected with either PEG10 gag-pol, gag, or nucleocapsid, and changes in gene expression were analyzed by RNA-seq. Transfection with PEG10 gag-pol induced the most gene expression changes, followed by a nucleocapsid (*Figure 6A*), with gag-transfected cells showing the fewest changes compared to the control (*Figure 6—source data 1*). Cluster profiling identified distinct groups of genes differentially regulated by specific PEG10 constructs. The first two groups consisted of genes that changed upon any type of PEG10 overexpression (*Figure 6B*), suggesting generalized responses to virus-like protein expression. One example of a gene upregulated by all forms of PEG10 expression was *TXNIP*, a regulator of oxidative stress, which is also elevated in multiple neurodegenerative conditions (*Tsubaki et al., 2020*; *Figure 6C*). The largest cluster profile (Group 3) consisted of genes

upregulated upon gag-pol and nucleocapsid transfection, but not gag transfection, highlighting the ability of the small nucleocapsid fragment to induce transcriptional changes in a manner similar to full-length gag-pol protein (*Figure 6A–B*).

Pathway analysis of differentially expressed genes also underscored the similarities in gene regulation between gag-pol and nucleocapsid expression. Gag-pol expression resulted in an overrepresentation of pathways including female-specific sex characteristics, consistent with the role of PEG10 in placental development (*Abed et al., 2019*; *Ono et al., 2006*), as well as those involved in axon extension and remodeling (*Figure 6—figure supplement 1A*). Nucleocapsid expression resulted in an even stronger overrepresentation of neuronal pathways, especially pathways involved in axon guidance and extension (*Figure 6D*). One notable example of an axon remodeling gene was *DCLK1*, which was significantly elevated in nucleocapsid and gag-pol, but unchanged in gag-expressing cells (*Figure 6E–F*, *Figure 6—source data 1*). Gag expression resulted in fewer transcript changes and did not alter neuronal gene expression, highlighting the unique effects of gag-pol and nucleocapsid (*Figure 6—figure supplement 1B*). Our data suggested a direct link between PEG10 abundance and changes in neuronal gene expression. To explore whether these transcripts are also altered in diseased neural tissue, we analyzed transcriptional data from a large cohort of post-mortem ALS patient spinal cord samples and observed similarly elevated levels of *TXNIP* and *DCLK1* transcripts (*Figure 6G*), suggesting that ALS involves similar pathways of transcriptional disturbance.

To better understand the transcriptional effects of PEG10 overexpression, splicing differences were examined across gag-pol, gag, and NC expression conditions. Consistent with changes to transcript abundance, both nucleocapsid and gag-pol expression resulted in splicing alteration of 150–200

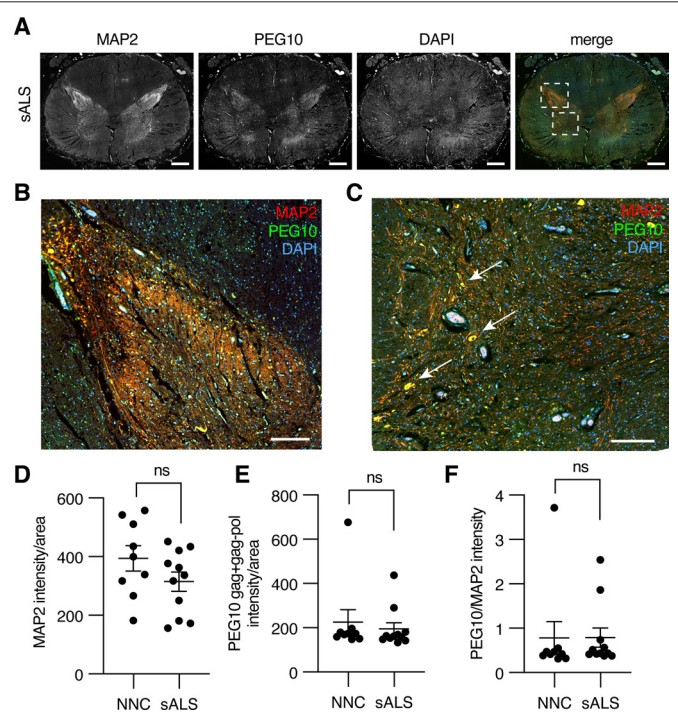

**Figure 7.** Paternally expressed gene 10 (PEG10) is enriched in horns of the lumbosacral spinal cord. (**A**) Tiled image of full-thickness lumbosacral spinal cord section demonstrating Microtubule-associated protein 2 (MAP2), PEG10, and DAPI staining. Merged image shows outlines of boxes for closer examination in (**B**) and (**C**). Scale bar 1 mm. (**B**) Merged image of the posterior horn of the spinal cord showing enrichment for both MAP2 and PEG10 signals. Scale bar 200 μm. (**C**) Merged image of the anterior horn of the spinal cord showing colocalization of MAP2 and PEG10 signal in cell bodies (arrows). Scale bar 200 μm. (**D–E**) Quantitation of (**D**) MAP2 and (**E**) PEG10 signal in spinal cord sections. Signal intensity of MAP2 and PEG10 were determined per mm$^2$ of spinal cord section for a cohort of non-neurological controls (NNC, n=9) and sporadic Amyotrophic Lateral Sclerosis (ALS) patients (sALS, n=11). (**F**) Relative PEG10 per MAP2 intensity was quantified for each section, to account for potential loss in neuronal staining due to ALS. No significant differences in intensity or localization were observed in the disease condition. For (**D–F**) mean ± SEM is shown.

genes, whereas gag had fewer effects (*Figure 6—source data 1*). However, there were no global changes to patterns of transcript splicing upon nucleocapsid expression (*Figure 6H*), nor were there global changes to mRNA trafficking (*Figure 6I–J*), indicating that the changes to transcriptional abundance are gene-specific or are unrelated to splicing and trafficking patterns.

## PEG10 gag-pol protein is elevated in human ALS tissues

To examine PEG10 abundance in the context of neurodegenerative disease, we performed immunofluorescence on fixed lumbosacral spinal cord tissue from ALS patients and matched controls. Microtubule-associated protein 2 (MAP2) was used to distinguish the white and gray matter of the spinal cord (*Figure 7A*). PEG10 staining was apparent in the gray matter of the spinal cord (*Figure 7A*), and was enriched in the posterior horn where somatosensory neurons are located (*Figure 7B*) as well as in what appear to be motor neuron bodies of the anterior horn (*Figure 7C*, arrows). To examine the abundance of PEG10 more closely, we compared the intensity of PEG10 signal in 11 sALS cases as compared to nine non-neurological controls (post-mortem samples from individuals who did not have any known neurological disease) and saw no dramatic differences in intensity or localization (*Figure 7D–F*).

A limitation of PEG10 immunofluorescence staining was a lack of antibody specificity: the anti-PEG10 antibodies tested all detected an epitope found in the gag region, meaning that immunofluorescence signal was generated from both gag and gag-pol forms of PEG10. Further, because the relative ratio

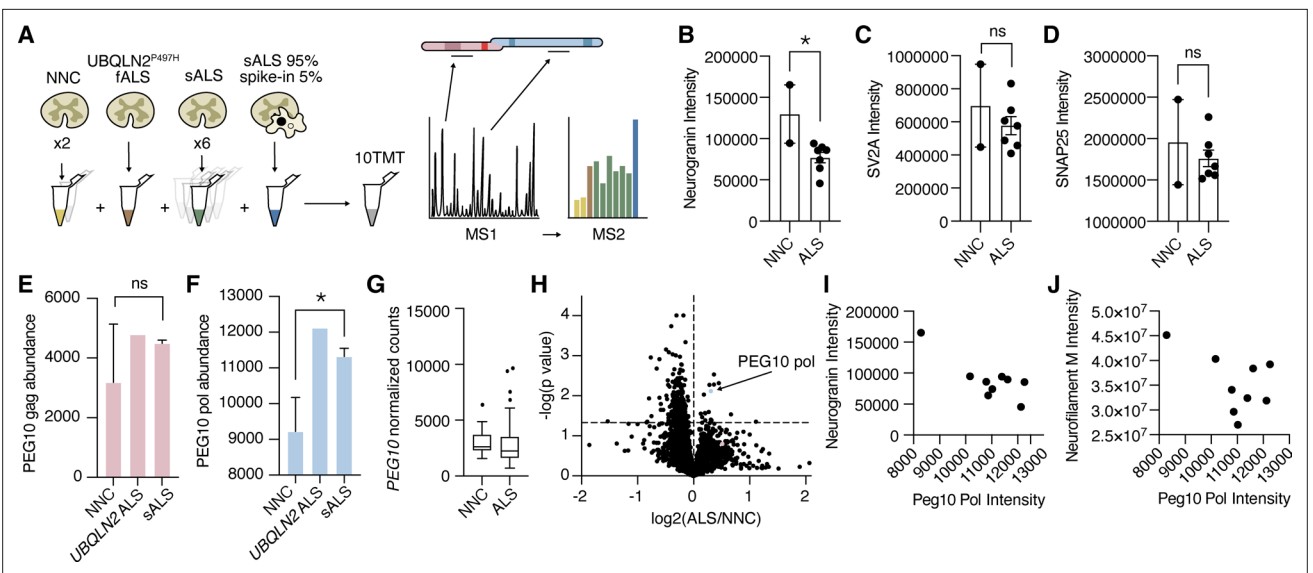

**Figure 8.** Paternally expressed gene 10 (PEG10) gag-pol protein accumulates in human Amyotrophic Lateral Sclerosis (ALS). (**A**) Schematic illustrating multiplexed global proteomic strategy to quantify PEG10 protein from the human lumbar spinal cord. Two non-neurological controls (NNC), one fALS case with a Ubiquilin 2 (*UBQLN2*) mutation, and six sporadic ALS cases were combined with a 'spike-in' PEG10 channel containing 5% lysate from cells transfected with HA-PEG10 gag-pol and 95% spinal cord lysate to normalize proteomic background complexity. All ten samples were labeled with tandem mass tags (TMT) and run as a 10-plex on LC-MS2. (**B–D**) Abundance of neurogranin (**B**), SV2A (**C**), and SNAP25 (**D**), in the human spinal cord. Mean ± SEM is shown and plotted for each marker. A significant decrease in neurogranin was detected between non-neurological controls NNC (n=2) and all ALS patients (n=7) by unpaired *t*-test (**E–F**) Abundance of PEG10 gag (**E**), and pol (**F**), in the human spinal cord. Mean ± SEM is shown. Significance was determined by Student's *t*-test. (**G**) RNA-Seq counts of *PEG10* from post-mortem lumbar spinal cord tissue of a larger cohort of patients with Classical ALS (n=127) or Non-Neurological Controls (NNC, n=17). Data are shown as 5–95% box and whisker plot and significance was determined by DESeq2. (**H**) Global proteomic analysis with 7465 individual proteins quantified. All ALS samples were grouped together, and two NNC were grouped to generate log₂ ratio of protein abundance and significance calculation by homoscedastic unpaired *t*-test. PEG10 pol is highlighted in blue, and PEG10 gag (not significant) is highlighted in pink. (**I**) Quantification of neurogranin intensity (*y*-axis) and PEG10 pol peptides' intensity (*x*-axis) are plotted for all ine spinal cord samples to demonstrate the relationship between the two markers. (**J**) Quantification of neurofilament medium intensity (*y*-axis) and PEG10 pol peptides' intensity (*x*-axis) are plotted for all nine spinal cord samples to demonstrate the relationship between the two markers. *p<0.05.

The online version of this article includes the following source data and figure supplement(s) for figure 8:

**Source data 1.** Proteomics results.

**Figure supplement 1.** Global proteomic analysis of paternally expressed gene 10 (PEG10) peptides.

of gag:gag-pol abundance has been estimated at ~60–80% gag (*Clark et al., 2007*; *Lux et al., 2010*), the majority of PEG10 signal is generated from gag protein. Because of the observed role of gag-pol in generating NC, we sought to specifically quantify the presence of gag-pol in human tissue. Unfixed post-mortem samples of the lumbosacral spinal cord were obtained for global proteomic analysis using a tandem mass tagging (TMT) approach for liquid-chromatography mass spectrometry (LC-MS). Samples were generated from two non-neurological controls, six sALS cases, and one *UBQLN2*[P497H]-mediated fALS case.

Traditional data-dependent proteomics, such as those used in previous PEG10 proteomic experiments (*Whiteley et al., 2021*), only fragment a subset of peptide ions as predetermined by the intensity of ions in MS1. As a result, the quantification of peptides favors those that are more highly abundant. PEG10 is expressed at very low levels in spinal cord lysate, and in our experience is often below the technical limit of detection by LC-MS. Therefore, we adapted a technique used for single-cell (*Budnik et al., 2018*) or small-cell population experiments (*Yi et al., 2019*) by adding a carrier, or 'spike-in' channel that triggers quantitation of PEG10 specifically (*Figure 8A*). The carrier channel, containing 95% spinal cord lysate and 5% cell lysate from HEK cells transfected with PEG10, drives the detection of PEG10 peptides by increasing the number of PEG10 peptide ions in MS1. The 'spike-in' channel is uniquely identified among spinal cord samples by labeling all samples analyzed in the experiment with a tandem mass tag (TMT) approach.

Overall, 7465 unique proteins were quantified across our samples. We observed changes in ALS tissues consistent with previously identified synaptic biomarkers of ALS and neurodegeneration, such as a significant reduction of neurogranin (*Kvartsberg et al., 2019*; *Vijayakumar et al., 2019*; *Figure 8B–D*, *Figure 8—source data 1*). Seven peptides were identified from the gag region of PEG10, and three peptides were identified from the pol region (*Figure 8—figure supplement 1*). While gag was not changed in *UBQLN2*-mediated or sporadic ALS samples (*Figure 8E*), peptides originating from PEG10 pol were significantly enriched in ALS compared to healthy controls (*Figure 8F*). This was specific to the protein level, as *PEG10* transcript counts were not elevated in ALS (*Figure 8G*). When PEG10 gag and pol were considered as unique proteins (two peptides per protein minimum, *Figure 8—source data 1*), PEG10 pol was among the most upregulated proteins in all ALS cases compared to healthy controls (*Figure 8H*). Its abundance in spinal cord tissue was also associated with a decrease in the intensity of neurogranin and neurofilament medium (*Figure 8I–J*), both of which are biomarkers for neurological disease (*Hellwig et al., 2015*; *Yuan and Nixon, 2021*). Taken together, the accumulation of PEG10 gag-pol in ALS tissue, paired with our findings of PEG10 self-cleavage and effects on gene expression, suggest that this pathway may represent a novel pathological contribution to the development of ALS with promise for novel drug development strategies.

## Discussion

Here, we have explored in detail the regulatory mechanisms and cellular consequences of PEG10 gag-pol self-processing and observed the accumulation of PEG10 gag-pol in both sporadic and *UBQLN2*-mediated ALS. PEG10 gag-pol exhibited protease-dependent self-cleavage and generated a nuclear-localized fragment that is reported to bind nucleic acids (*Abed et al., 2019*; *Segel et al., 2021*; *Steplewski et al., 1998*). This fragment was sufficient to change transcript abundance in the cell, including the expression of genes involved in axon remodeling, linking PEG10 dysregulation to neuronal dysfunction. Finally, we observed a specific accumulation of PEG10 gag-pol in the spinal cord tissue of ALS patients, suggesting that PEG10 activity may contribute to disease in humans.

Data outlined here suggest a specific relationship between UBQLN2 and the domesticated retrotransposon, PEG10. Notably, only UBQLN2 is capable of restraining PEG10 abundance, meaning that Ubiquilins do not universally share client populations. The unique ability of UBQLN2 to mediate PEG10 degradation is likely dependent on the PXX domain of UBQLN2, which is not found in other *UBQLN* genes, and is a mutational hotspot in *UBQLN2*-mediated fALS (*Deng et al., 2011*). Rescue of UBQLN TKO cells with either *UBQLN2*[P506T] or *UBQLN2*[P497H] resulted in changes to gag-pol abundance, which is consistent with our hypothesis. Interestingly, we observed that *UBQLN2*[P497H] expression was sufficient to control endogenous PEG10 levels whereas *UBQLN2*[P506T] was not. However, both alleles are deficient compared to WT *UBQLN2* when cells overexpress tagged PEG10. Together, these data indicate that there are allele-specific differences in the ability of UBQLN2 to facilitate PEG10 degradation. Different UBQLN2 alleles have been shown to display unique properties of phase separation

(*Dao et al., 2019*), autophagy (*Wu et al., 2020*), and protein degradation (*Chang and Monteiro, 2015*), which may explain their unique behavior with regard to PEG10. Regardless, in the human spinal cord of an ALS patient bearing the *UBQLN2^{P497H}* allele, PEG10 pol protein was elevated compared to controls.

It is also notable that only the gag-pol form of PEG10 is a client of UBQLN2, due in part to the presence of the unique PPR of the pol domain. *UBQLN2* and the PPR of PEG10 exist only in eutherian mammals, suggesting an evolutionary relationship. In comparison, the rest of the *UBQLN* gene family is more conserved among eukaryotes: *UBQLN3, 5*, and *L* are shared among earlier mammalian ancestors, *UBQLN1* first appears in fish species, and *UBQLN4* is thought to reflect adaptation of the ancestral *Dsk2* gene in single-celled eukaryotes (*Marín, 2014*). These points, paired with the finding that PEG10 was by far the most dysregulated protein in global proteomic studies of *UBQLN2* loss (*Whiteley et al., 2021*), raise the intriguing possibility that contrary to its presumed role as a regulator of generalized protein degradation, *UBQLN2* may have evolutionarily arisen specifically to restrain PEG10 abundance. Further, the unique relationship between *PEG10* and *UBQLN2* highlights a potential genetic conflict between the requirement for PEG10 expression in placental development (*Ono et al., 2006*) with pathological roles in neural tissue which would have necessitated the evolutionary development of a tissue-specific inhibitor of gag-pol activities. In addition to our findings in ALS, PEG10 abundance is also linked to the neurodevelopmental disorder Angelman's syndrome (*Pandya et al., 2021*). This observed dysregulation of PEG10 in affected tissues of neurological disease, as well as the unique enrichment of UBQLN2 in neuromuscular tissues, further supports our hypothesis.

Proteasomal degradation is a major contributor to the UBQLN2-mediated PEG10 phenotype. Previously, *UBQLN2* loss in neuronal cultures was shown to delay PEG10 gag-pol degradation (*Whiteley et al., 2021*). However, it remains unknown whether ubiquitination of PEG10 is necessary for UBQLN2-mediated degradation via the proteasome. Canonically, Ubiquilins have been thought to work with one or more E3 ubiquitin ligases to facilitate ubiquitination and degradation of client proteins (*Itakura et al., 2016*); however, the specific E3(s) that Ubiquilins work with remain unidentified. In this case, E3 ligases known to facilitate the suppression of retroelements would be prime candidates to examine. In fact, overexpression of the yeast E3 ligase SCF complex subunit Hrt1 (also known as Rbx1) potently inhibits Ty3 retrotransposition (*Seol et al., 1999*), though it is unknown whether inhibition is dependent on degradation of Ty3 protein.

Our work also adds insight into the biology of PEG10. Previous work has suggested that PEG10 self-processes in a manner reminiscent of the self-processing of retrotransposons (*Clark et al., 2007*; *Golda et al., 2020*; *Lux et al., 2005*). Our study confirms and adds to these findings by determining precise locations of self-cleavage, the fate of cleaved products, and the effects of these cleaved products on cellular health. PEG10 behaves similarly to its ancestor, the Ty3 retrotransposon, in its ability to generate a short 'NC' protein fragment consisting of the isolated zinc finger (*Kirchner and Sandmeyer, 1993*), which uniquely moves to the nucleus (*Beliakova-Bethell et al., 2009*). Surprisingly, we found that expression of just the PEG10 NC cleavage product resulted in unique transcriptional changes to the cell. Neuronal genes were some of the most altered transcripts upon gag-pol and nucleocapsid transfection, implicating a link between protease-dependent PEG10 self-cleavage, nucleocapsid liberation, and neuronal dysfunction. However, further work is needed to identify the molecular mechanism by which nucleocapsid mediates transcriptional changes. The PEG10 zinc finger has been reported to bind cellular mRNAs (*Abed et al., 2019*; *Segel et al., 2021*), as well as DNA (*Steplewski et al., 1998*). Therefore, nucleocapsids may influence gene expression by acting as a classical transcription factor, or as a regulator of transcript abundance, stability, or availability. While we did not observe bulk changes to trafficking or splicing, PEG10 may regulate mRNA stability, trafficking, or splicing in a gene-dependent manner. These possibilities remain to be elucidated.

Strikingly, among the more than 7000 proteins quantified by our analysis, PEG10 pol was the fifth most significantly upregulated protein in spinal cord tissue of ALS patients, suggesting a strong connection to disease and the potential utility of PEG10 as a novel biomarker for ALS progression. Even in the absence of overt *UBQLN2* mutation, PEG10 gag-pol protein is accumulated in post-mortem tissue of sALS patients compared to non-neurological controls. We hypothesize that this disease-wide accumulation is related to the inclusion of UBQLN2 in sALS-associated protein aggregates (*Deng et al., 2011*; *Williams et al., 2012*), which would lead to a loss of function phenotype despite normal *UBQLN2* expression. This further suggests that PEG10 gag-pol accumulation may be

considered alongside TDP43, FUS, and UBQLN2 mislocalization and aggregation as shared hallmarks of familial and sporadic ALS (**Blokhuis et al., 2013**).

Elevated PEG10 has been observed in Angelman's syndrome, where its overexpression has been shown to influence neuronal trafficking in the embryonic brain (**Pandya et al., 2021**). In line with these findings, our work supports a putative role for PEG10 gag-pol accumulation as a mechanism of ALS disease progression. The ability of elevated PEG10 to influence gene expression in cell culture models suggests that its accumulation in human tissues may induce similar disruption to transcriptional programming. Indeed, elevated PEG10 protein abundance, as well as transcriptional changes to the genes *TXNIP* and *DCLK1*, were observed in both PEG10-expressing cell lines and in ALS tissue compared to controls.

Additional studies are required to determine the precise contribution of PEG10 to the transcriptional changes observed in ALS tissues and to in vivo neurodegeneration. Ultimately, further understanding of PEG10 biology in the context of ALS, as well as other conditions where PEG10 is elevated, may provide novel avenues for therapeutic development.

# Materials and methods

## Key resources table

| Reagent type (species) or resource | Designation | Source or reference | Identifiers | Additional information |
|---|---|---|---|---|
| strain, strain background (*Escherichia coli*) | Rosetta | Sigma | 70954–3 | |
| strain, strain background (*Escherichia coli*) | DH5α | Fisher Scientific | 18265017 | |
| cell line (*Homo sapiens*) | HEK293 WT and *UBQLN1, 2, 4* TKO cells, inducible *UBQLN2* expression | **Itakura et al., 2016** | | |
| cell line (*Homo sapiens*) | Human ESCs (H9) WT and *UBQLN1, 2,* or *4* sinbgle KO cells | Harvard Medical School Cell Biology Initiative for Genome Editing and Neurodegeneration | | |
| antibody | anti-UBQLN1/2 (Mouse Monoclonal M03) | Abnova | H00029978; AB_627374 | WB (1:1000) |
| antibody | anti-UBQLN4 (Rabbit Polyclonal) | GeneTex | GTX85267; AB_10725990 | WB (1:1000) |
| antibody | anti-Myc tab (Mouse Monoclonal GT002) | Sigma | SAB2702192 | WB (1:2000) |
| antibody | anti-PEG10 (Rabbit Polyclonal) | Proteintech | 14412–1-AP; AB_10694427 | WB (1:1000) |
| antibody | anti-HA (Mouse Monoclonal HA-7) | Sigma | H3663; AB_262051 | WB, IF (1:1000) |
| antibody | anti-NC (Rabbit Polyclonal) | Thermo Fisher (custom) | n/a | WB (1:1000) |
| antibody | anti-tubulin (Mouse Monoclonal DM1A) | Novus | NB100-690; AB_521686 | WB (1:20000) |
| antibody | anti-MAP2 (Chicken Polyclonal 28225) | Biolegend | 822501; AB_2564858 | IF (1:2000) |
| antibody | α-mouse IgG 680 (Goat Polyclonal) | Licor | 926–68070; AB_10956588 | WB (1:20000) |
| antibody | α-mouse IgG 800 (Goat Polyclonal) | Licor | 926–32210; AB_621842 | WB (1:20000) |
| antibody | α-rabbit IgG 680 (Goat Polyclonal) | Licor | 926–68071; AB_10956166 | WB (1:20000) |
| antibody | α-rabbit IgG 800 (Goat Polyclonal) | Licor | 926–32211; AB_621843 | WB (1:20000) |

*Continued on next page*

*Continued*

| Reagent type (species) or resource | Designation | Source or reference | Identifiers | Additional information |
|---|---|---|---|---|
| antibody | Alexa568 α-mouse (Goat Polyclonal) | Invitrogen | A11004; AB_2534072 | IF (1:300) |
| antibody | Alexa568 α-chicken (Goat Polyclonal) | Invitrogen | A11041; AB_2534098 | IF (1:300) |
| antibody | Alexa647 α-rabbit (Goat Polyclonal) | Invitrogen | A32733; AB_AB2633282 | IF (1:300) |
| Transfected construct (*Saccharomyces cerevisiae*) | LIFEACT (Abp140 AA1-17) | GenBank; *Riedl et al., 2008* | n/a | pCDNA3.1 |
| Transfected construct (*Homo sapiens*) | gag-pol; PEG10 (AA1-708) | GenBank | NP_055883.2 | pCDNA3.1 |
| Transfected construct (*Homo sapiens*) | gag-pol$^{ASG}$;PEG10 (AA1-708)* D370A | GenBank | NP_055883.2 | pCDNA3.1 |
| Transfected construct (*Homo sapiens*) | Gag; PEG10 (AA1-325) | GenBank | NP_001035242.1 | pCDNA3.1 |
| Transfected construct (*Homo sapiens*) | CA; PEG10 (AA1-259) | GenBank | NP_001035242.1 | pCDNA3.1 |
| Transfected construct (*Homo sapiens*) | NC; PEG10 (AA260-325) | GenBank | NP_001035242.1 | pCDNA3.1 |
| Transfected construct (*Homo sapiens*) | Tag-less NC; PEG10 (AA260-325) | GenBank | NP_001035242.1 | pCDNA3.1 |
| Transfected construct (*Homo sapiens*) | gag111; PEG10(AA1-111) | GenBank | NP_001035242.1 | pCDNA3.1 |
| Transfected construct (*Mus musculus*) | gag-pol; PEG10 (AA1-1006) | GenBank | NP_570947.2 | pCDNA3.1 |
| Transfected construct (*Mus musculus*) | gag-pol$^{ASG}$;PEG10 (AA1-1006)* D420A | GenBank | NP_570947.2 | pCDNA3.1 |
| Transfected construct (*Homo sapiens*) | gag-pol Dendra2; PEG10 (AA1-708)-Dendra2 | GenBank | NP_055883.2 | pDendra2 |
| Transfected construct (*Homo sapiens*) | gag-pol$^{ASG}$ Dendra2; PEG10 (AA1-708)* D370A-Dendra2 | GenBank | NP_055883.2 | pDendra2 |
| Transfected construct (*Homo sapiens*) | ΔPPR Dendra2; PEG10 (AA1-681)-Dendra2 | GenBank | NP_055883.2 | pDendra2 |
| Transfected construct (*Macaca Mulatta*) | gag-pol Dendra2; PEG10 (AA1-742)-Dendra2 | GenBank | NP_001165893.2 | pDendra2 |
| Transfected construct (*Bos taurus*) | gag-pol Dendra2; PEG10 (AA1-788)-Dendra2 | GenBank | NP_001120682.1 | pDendra2 |
| Transfected construct (*Mus musculus*) | gag-pol Dendra2; PEG10 (AA1-1006)-Dendra2 | GenBank | NP_570947.2 | pDendra2 |
| Transfected construct (*Phascolarctos cinereus*) | gag-pol Dendra2; PEG10 (AA1-624)-Dendra2 | GenBank | XM_021000084.1 (+downstream CDS continuation to *pol* stop codon) | pDendra2 |
| Transfected construct (*Phascolarctos cinereus*) | gag-pol Dendra2+HsPPR; Pc PEG10 (AA1-624)+Hs PEG10 (AA682-708) Dendra2 | GenBank | XM_021000084.1 (+downstream CDS continuation to *pol* stop codon) And NP_055883.2 | pDendra2 |
| Transfected construct (*Homo sapiens*) | 6XHIS-SUMO2-NC; 6 X His tag + Sumo2 tag+PEG10 (AA260-325) | GenBank | NP_001035242.1 | pETSUMO2 |
| chemical compound, drug | cycloheximide | Sigma | 01810 | |

| Reagent type (species) or resource | Designation | Source or reference | Identifiers | Additional information |
|---|---|---|---|---|
| chemical compound, drug | Bortezomib | MP Biomedicals | IC0218385905 | |
| other | Prolong Gold DAPI anti-fade mounting media | Invitrogen | P36941 | Mounting medium for coverslips |

## Custom antibody generation

The custom antibody used to detect the isolated NC fragment was generated via ThermoFisher. PEG10 Nucleocapsid (NC) was cloned into pET-SUMO2, and the resulting recombinant plasmid was transformed in Rosetta cells. A single colony was used to inoculate Luria broth (LB) with carbenicillin (100 µg/mL, Gold Biotechnology) and chloramphenicol (20 µg/mL, Gold Biotechnology) and grown overnight with shaking at 37 °C. Following this initial growth, 5–10 mL of the culture was used to inoculate 750 mL of LB supplemented with the same antibiotics and incubated at 37 °C with shaking until an $OD_{600}$ = 0.6–0.8 was achieved. Expression was induced by the addition of 0.1 M IPTG and cultures were left shaking overnight at 16 °C. Cells were harvested by pelleting at 4500 × g for 30 min at 4 °C.

The resulting pellet was resuspended in 30 mL of Lysis Buffer (50 mM Tris-HCl, 500 mM NaCl, 10% glycerol, 10 mM imidazole, 0.1 mM DTT, 1 mM PMSF, (pH 7.4) and one protease inhibitor cocktail tablet (Sigma Aldrich)).

The resuspended cells were lysed by sonication for 15 min with an amplitude of 80, in 30 s on/off intervals. The lysed cells were then centrifuged at 14,000 × g at 4 °C for 1 hr. The supernatant was then processed by immobilized metal affinity chromatography (IMAC) using nickel-NTA resin (Gold Biotechnology) in a gravity column, equilibrated with Lysis Buffer. Cleared lysate was added to the column and flowthrough was collected. The resin was then washed four times with 20 mM sodium phosphate, 500 mM NaCl, 20 mM imidazole (pH 7.4), and flowthrough from each wash was collected. Protein was then eluted with ~30 mL of Elution Buffer (20 mM sodium phosphate, 500 mM NaCl, 500 mM imidazole (pH 7.4)). The presence of SUMO-NC in each fraction was determined by SDS-PAGE and Coomassie staining (ThermoFisher). Fractions containing SUMO-NC were placed in dialysis tubing (Pierce) and dialyzed in Dialysis Buffer (20 mM sodium phosphate, 100 mM NaCl, and 0.1 mM DTT). Dialyzed sample was analyzed by Bradford assay (ThermoFisher) to determine relative protein amounts, and ULP1 protease (1.25 mg/mL) was added at 1:100 ratio and incubated overnight at 4 °C. The dialyzed sample was brought to 1 M NaCl and 20 mM imidazole, and processed by a subsequent round of IMAC. Nickel-NTA resin was equilibrated in 20 mM sodium phosphate, 1 M NaCl, and 20 mM imidazole (pH 7.4). The sample was then added to the column and washed three times with 20 mM sodium phosphate, 1 M NaCl, and 20 mM imidazole. Dialysis was then performed again, overnight at 4 °C, using 3000 MWCO dialysis tubing (Pierce).

The dialyzed sample was subjected to another round of IMAC, with nickel-NTA resin equilibrated with 20 mM sodium phosphate and 100 mM NaCl (pH 7.4). Bound protein was then washed with 20 mM sodium phosphate and 100 mM NaCl (pH 7.4). NC was found to bind nickel-resin due to the contained zinc-finger, and was eluted from the column using Elution Buffer, and all fractions were collected and analyzed by SDS-PAGE followed by Coomassie staining. Purified protein was then dialyzed in Dialysis Buffer without DTT, using 3000 MWCO dialysis tubing, and concentrated to 1 mg/mL using a 3000 MWCO concentrator (Millipore). Concentrated protein was snap frozen and stored at –80 °C before shipment to ThermoFisher for antibody production.

Antibody generation via ThermoFisher involved conjugation to KLH immunogen and subsequent immunization of rabbits. Rabbits were immunized and boosted three times with protein. Rabbits were bled at 72 days and antibody reactivity was confirmed by ELISA. After ELISA confirmation, the antibody from the collected serum was column purified for downstream use.

## Cell lines

WT HEK293 cells and HEK cells lacking *UBQLNs* 1,2, and 4 ('TKO') were a gift from Dr. Ramanujan Hegde of the Medical Research Council Laboratory of Molecular Biology. Their identity was authenticated by examination of morphology, Ubiquilins 1, 2, and 4 expressions, and PEG10 expression by western blot. They are tested annually for mycoplasma contamination. WT and TKO HEK293 cells

were cultured in Dulbecco's modified Eagle's medium (Gibco, cat #12800–082) supplemented with 1% penicillin/streptomycin (Gibco, cat #15140163), 1% L-glutamine (R&D Systems, Inc, cat #R90210), and 10% FBS (Peak Serum, cat #PS-FB3). HEK293 TKO cells stably transfected with doxycycline- (dox-) inducible constructs expressing *Myc-UBQLN2*, or *Myc-UBQLN2^{P497H}* or *Myc-UBQLN2^{P506T}* were provided by Dr. Hegde via Dr. Miguel Prado and are described in Itakura et al. Dox-inducible cells were cultured in DMEM (Gibco, cat #12800–082) supplemented with 1% penicillin/streptomycin (Gibco, cat #15140163), 1% L-glutamine (R&D Systems, Inc, cat #R90210), and 10% tet-system approved FBS (Gibco, cat #A4736301), as contaminating doxycycline in standard FBS was sufficient to stimulate high levels of *UBQLN2* expression. Leaky expression of *Myc-UBQLN2* constructs in a dox-free medium resulted in endogenous levels of tagged UBQLN2 expression.

Human H9 hESCs lacking either *UBQLN1*, *2*, or *4*, were generated at the Harvard Medical School Cell Biology Initiative for Genome Editing and Neurodegeneration according to *Whiteley et al., 2021*. They were verified as mycoplasma-free upon shipment to CU Boulder. Their identity was authenticated by examination of morphology upon passaging, and expression of Ubiquilins 1, 2, and 4. hESCs were cultured in either E8 (Gibco, cat #A1517001) or TeSR-E8 medium (StemCell Technologies, cat #05990) on six-well tissue culture plates with Matrigel (Corning, cat #354277, Lot #0048006). Medium was changed daily. Cells were passaged by treatment in 0.5 mM EDTA (Sigma Aldrich, cat #E5134-500G) in sterile D-PBS (Gibco, cat #21600069) and replated in media at an approximate 1:6 dilution. The remaining non-passaged cells were washed in D-PBS three times and pelleted at 300 × g for analysis.

## Cell transfection

WT or TKO HEK293 cells were grown to 70% confluency in 12-well plates and transfected with 1 µg plasmid DNA in Lipofectamine 2000 (Invitrogen, cat #11668027) and Opti-Mem medium (Gibco, cat #11058021), according to manufacturer's instructions. After 48 hr, cells were harvested for western blot, qPCR, RNA-seq, or immunofluorescence. For protein degradation studies, cells were treated with 100 µg/mL cycloheximide (Sigma) starting at 24 hr post-transfection.

## Human tissue samples

Human tissue samples were acquired from the Target ALS Multicenter Human Postmortem Tissue Core. Fixed, paraffin-embedded full-thickness sections of the lumbosacral spinal cord were obtained from 11 sporadic ALS patients and nine non-neurological controls. Samples were from a mixture of males and females.

Unfixed, full-thickness sections of the spinal cord from the lumbosacral region were obtained from two non-neurological controls, one ALS patient with a pathogenic *UBQLN2* mutation, and seven sporadic ALS cases. All cases were from females. More detail can be found in *Figure 8—source data 1*.

## Cloning

All constructs were designed with a CMV promoter using Gibson or restriction cloning (see list of constructs) and transformed into chemically competent DH5α *E. coli* cells (Invitrogen). Transformed *E. coli* were plated on either 50 µg/mL kanamycin (Teknova, cat #K2151) or 100 µg/mL carbenicillin (Gold Biotechnology, cat #C-103–5) LB agar (Teknova, cat #L9115) plates overnight at 37 °C. Single colonies were picked and grown overnight in 5 mL LB Broth (Alfa Aesar, cat #AAJ75854A1) with kanamycin or carbenicillin at 37 °C with shaking at 220 rpm. The following day, shaking cultures were mini-prepped (Zymo, cat #D4212) and sent for Sanger Sequencing (Azenta). Sequence-verified plasmids were then midi-prepped (Zymo, cat #D4201) for use in transfection.

## Flow cytometry

WT and TKO HEK293 cells were transfected in 96-well plates. Cells were harvested 48 hr after transfection in FACS Buffer (D-PBS (Gibco, cat #21600069), 2% FBS (Peak Serum, cat #PS-FB3), 0.1% Sodium Azide (Millipore Sigma, cat #26628-22-8)), and analyzed on a BD FACSCelesta. Triplicate wells were transfected within a plate to serve as technical replicates, and experiments were performed four independent times. FlowJo software was used for data analysis.

Cells were first gated in the FSC-A vs. SSC-A using the polygon gating tool. Within the 'cells' population, CFP-positive cells were gated on 405 nm vs. SSC-A. The Dendra2 Green/CFP parameter

was created by deriving a novel parameter of the 488 references by the 405 references, and making a logarithmic scale with a minimum of 0.0001 and a maximum of 10. The geometric mean of the custom Dendra2 Green/CFP parameter from the CFP positive population was exported and used to generate graphs. For Dendra2 Green/CFP from HEK cell UBQLN2, rescue experiments in *Figure 2*, biological replicates from experiments performed on different days were normalized based on a 'day average' value for all samples.

## Western blotting

Cell pellets were collected by centrifugation, washed in PBS, and lysed in urea buffer (8 M urea (Fisher Chemical, cat #U153), 75 mM NaCl (Honeywell Fluka, cat #6003219), 50 mM HEPES (Millipore Sigma, cat #H3375) pH 8.5, 1 x tab cOmplete Mini EDTA-free protease inhibitor cocktail tablet (Roche, cat #11836170001)). Lysate was centrifuged for 10 min at 21,300 × g and the supernatant was collected.

Protein was quantified by BCA (Pierce, cat #23227) and 1 x Laemmli sample buffer supplemented with βME (Sigma Aldrich, cat #M3148) was added to samples before SDS-PAGE. Samples were run in NuPage MES Running Buffer (Invitrogen, cat #NP000202) on a 4 to 12% NuPage Bis-Tris gel (Invitrogen, cat #NP0321) and wet transferred on nitrocellulose membrane (Amersham Protran, cat #10-6000-14) for either 90 min at 100 V on ice (BioRad, cat #1703930) or 60 min at 10 V (Invitrogen, cat #NW2000).

Membranes were blocked using 1:1 LICOR blocking buffer (cat #927–70001) and 1 x TBS (50 mM Tris-Cl (MP Biomedicals, cat #MP04816100) pH 7.4, 150 mM NaCl), for 30 min at room temperature. Membranes were incubated in primary antibody overnight at 4 °C and washed in 1 x TBST (1 x TBS, 0.1% Tween VWR, cat #M147-1L) in 3-5 min intervals. Membranes were then incubated in LICOR secondary antibody for 30 min in the dark. After 3 x more washes, banding patterns were visualized using LICOR Odyssey CLx and data analysis was performed using LICOR ImageStudio Software. Each protein quantification was normalized to the average intensity across all samples in each replicate western blot to correct for technical variation across experiments.

## Virus-like particle isolation

HEK293 cells (RRID: CVCL_0045) were plated in a six-well plate at a density of $4 \times 10^5$ cells per well. 24 hr after plating, cells were transfected and media was replaced 6 hr later. Cultured media was harvested 48 hr after transfection and pre-cleared by centrifugation at 2000 × g for 15 min at 4 °C. In parallel, cell lysate was collected for western blot as previously described. The VLP fraction was isolated by ultracentrifugation (Beckman Coulter L8-70M Preparative Ultracentrifuge) at 134,000 × g for 4 hr with a 30% sucrose cushion. After ultracentrifugation, media and sucrose were aspirated, and the VLP-containing pellet was resuspended in an 8 M urea lysis buffer. VLP production was analyzed by western blot.

## Phylogenetic alignment of PEG10

PEG10 protein sequences were curated from NCBI for selected eutherian mammals and manually curated from marsupial mRNA sequences from NCBI due to a lack of automatically annotated frame-shifting sites. Sequences were then aligned with Geneious 3.0 using a MUSCLE-based algorithm with 8 iterations. Alignment was visualized with unique colors for each amino acid.

## Structure prediction

Structure prediction for the PEG10 gag protein (AA1-325) was performed using the Phyre 2.0 webserver (*Kelley et al., 2015*) using the intensive modeling mode. 243 of 325 amino acids were modeled with >90% confidence, with amino acids 89–314 predicted with confidence >99% against reference structures including *Saccharomyces* Ty3 (PDB 6R24), *Drosophila* and *Rattus* Arc (PDB 6TAR, 6TAQ, and 6GSE), HIV (PDB 6RWG), and a partial structure of *Homo sapiens* PEG10 gag (PDB 7LGA). The predicted PEG10 structure was visualized using UCSF Chimera.

## Immunofluorescence

WT HEK293 cells were either plated from 24-well plates onto Alcian blue-treated (Newcomer Supply, cat #1002 A) round coverslips (Electron Microscopy Sciences, cat #7223101SP) and transfected for 48 hr before harvesting, or were plated 24 post-transfection onto coverslips and harvested for staining

after 24 hr. Coverslips were fixed at the time of harvest in 4% PFA (Thermo Scientific, cat #28906). Cells were then either submerged in 1% PFA for overnight storage or washed three times in 1 x PBS. Cells were permeabilized in 0.25% Triton-X (Sigma Aldrich, cat #X100) in PBS and incubated in blocking buffer (7.5% BSA (Gibco, cat #15260037) diluted to 5% in PBS, 0.1% tween) for 30 min. Cells were incubated for 1 hr in primary antibody before three 5 min washes in 1 x PBS-T (0.1% Tween in PBS). Cells were then incubated in secondary antibodies for 1 hr in the dark. Cells underwent three more 5 min 1 x PBS-T washes and were then rinsed three times in DEPC water. After sufficient drying, 5–10 mL of Prolong Gold DAPI anti-fade mounting media (Invitrogen, cat #P36941) was added to coverslips. Coverslips were then mounted on clear microscope slides and cured overnight in the dark at room temperature before imaging.

For staining of the human spinal cord, sections of paraffin-embedded, fixed lumbosacral spinal cord tissue obtained from Target ALS were de-paraffinized and heat-induced antigen retrieval was performed by incubating slides in sodium citrate at 90 °C for 20 min. Sections were incubated overnight in primary antibodies. The following day samples were washed 3 x with PBS containing 0.1% Tween (PBS-T) and stained with secondary antibody for 1 hr in the dark at room temperature followed by three washes with PBS-T and one wash with DEPC water. Coverslips were mounted using Prolong Gold DAPI anti-fade mounting media and cured overnight prior to imaging. Microscopy was performed using a Nikon Widefield microscope maintained by the BioFrontiers Advanced Light Microscopy Core using a 10 x objective and NIS Elements software.

### Oligo dT fluorescence in situ hybridization

Transfected WT HEK293 cells were cultured on round coverslips (Electron Microscopy Sciences, cat #7223101SP) and hybridized according to Stellaris protocol for hybridization of adherent cells (Biosearch Technologies). A T30 Poly A probe (Stellaris, Biosearch Technologies, Custom oligo) was used to detect polyA mRNA tails as a measure of total mRNA by cellular compartment. 10 images were obtained for each transfection condition and randomly assigned image names for blind quantification. Distinct single cells were quantified using FIJI software XOR function to quantify the mean signal intensity of the nucleus and cytoplasm. The nucleus to cytoplasmic ratio was calculated for a minimum of 60 cells for each condition.

### Microscopy

Confocal microscopy was performed on a Nikon AR1 LSM confocal microscope maintained by the BioFrontiers Advanced Light Microscopy Core using a 20 x Air objective and NIS Elements Nikon software. Acquisition intensity and pinhole size were fixed across samples to control for signal intensity and variability. For visualization purposes only, the image intensity of visualized channels was increased from acquisition parameters according to FIJI software parameters.

For imaging of human spinal cord sections, slides were imaged using a Nikon widefield microscope maintained by the BioFrontiers Advanced Light Microscopy Core with a 10 x objective and NIS Elements Nikon software. Acquisition intensity was fixed across samples. The fluorescence images were analyzed using home-built code in the MATLAB (R2021b) language. Due to the large file sizes of the raw data, the images were loaded in quadrants, and each quadrant was reduced in size by calculating the average intensity of 2 × 2 pixel blocks. The reduced images were then reconstituted and saved as uncompressed TIFF files. These reduced files were used in the rest of the analysis. We validated that this approach did not lead to major differences in the final data by comparing the mean intensity measured using a full-sized image and the mean intensity of our reduced version. For each image, a binary mask to identify regions of interest was generated using thresholding the DAPI-stained nuclear image. Manual corrections were made as necessary to remove erroneously identified regions, e.g., pieces of detached tissue. The average fluorescence intensities of MAP2 and PEG10 signals within the masked region were then calculated.

### Sample preparation for RNA sequencing

HEK293 cells were grown in 12-well plates, transfected for overexpression of genes of interest, and collected for RNA isolation 48 hr later. Cells were pelleted and RNA was extracted using the RNEasy Mini Kit (Qiagen, cat #74106) with on-column DNAse digestion (Qiagen, 79256). Isolated RNA was quantified and quality controlled by nanodrop (Thermo Fisher), concentration was normalized, and samples were stored at –80 °C.

## RNA-sequencing analysis

Poly A Selected Total RNA Library paired-end sequencing was performed at Anschutz Medical Campus on an Illumina NovaSEQ 6000. Sequencing produced between 24–104 million filtered paired-end reads across all samples. Quality of reads was determined using FastQC (the average reads/base quality for all samples in the lane was at least 88% ≥Q30) and reads were mapped to GRCh38.p13 (*Frankish et al., 2019*) using STAR version 2.7.3 (*Dobin et al., 2013*). STAR alignment.bam files were indexed and sorted before count matrix generation using Samtools 1.8 and the featureCounts software package (*Li et al., 2009*).

Count files were converted to readable format in unix and imported into Rstudio for DESeq2 analysis using R. Data were quality controlled by estimating size factors and genewise dispersion estimates for variance in gene expression. Shrinking was used to fit dispersion curves and principal component analyses dictated design parameters for differential gene expression analysis. Gene expression patterns were tracked using DESeq2 (*Love et al., 2014*) using harvest date and transfection construct as major variables, as well as Cluster Profiling (*Yu et al., 2012*), and GO Term expression (*Luo and Brouwer, 2013*; *Yu et al., 2015*) analyses. Significance of gene expression changes was determined with a p-adjusted cutoff of .05. Gene groups were determined with DEGReport (*Pantano, 2021*) using a reduced cluster model in which outliers of cluster distribution were removed.

Pathway analysis was performed using the enrichGO program (*Yu et al., 2015*) on all GO-term pathways with a $log_2$foldchange cutoff of 0.5 and a p-value of 0.05 of significantly changed genes for each pairwise analysis (pCDNA negative control vs. gag-pol, vs. gag, and vs. NC). The top five pathways by p-value were visualized.

Splicing analysis was performed using the MAJIQ Quantifier followed by the MAJIQ Builder to determine differentially spliced genes (*Vaquero-Garcia et al., 2016*), and visualized using the MAJIQ Voila Viewer with a $\Delta\phi$ threshold of 0.1 and significance of 0.05. Splice variant classification analysis was performed using the MAJIQ classifier (*Vaquero-Garcia et al., 2021*) with permission and assistance from Dr. Yoseph Barash.

## Target ALS dataset RNA-seq analysis

Raw RNA-seq reads from the lumbar spinal cord of the Target ALS: New York Genome Center dataset was obtained; at the time of analysis, this dataset included 127 Classical/Typical ALS cases, 17 non-neurological controls (deceased donors who did not exhibit neurological disease), and two known *UBQLN2*-mediated cases. Classical/Typical ALS refers to the process of the disease course and is comprised of a majority of sporadic ALS cases unless specifically noted to have the sequenced mutation of known ALS susceptibility genes, of which *UBQLN2* was included. Approximately half of the samples were male/female. Reads were aligned to the human genome (hg38) using STAR version 2.5.2b as above (*Dobin et al., 2013*), and analyzed in Rstudio with DESeq2 (*Love et al., 2014*) including sex and 'Subject.Group.Subcategory' (disease type) as major variables. Significance of gene expression changes was determined with a p-adjusted cutoff of 0.05, and normalized counts was used for the visualization of target genes.

## Sample preparation for mass spectrometry analysis

Human spinal cord samples from two non-neurological controls, six sporadic, classical ALS cases, and one case of *UBQLN2*-mediated fALS were first sectioned on a cryostat (Leica) to ensure even tissue representation of protein samples. Ten to twenty 15 μm-thickness sections from each patient were homogenized in 8 M urea lysis buffer, the lysate was spun at 15,000 rpm for 15 min at 4 °C to remove insoluble material, and supernatant protein content was quantified by BCA analysis (Pierce, cat #23227). Separately, HEK cells were transfected with *Homo sapiens* HA-PEG10 gag-pol, lysed 48 hr later, and mixed in a 95:5 ratio of sALS spinal cord lysate to HEK cell lysate. Approximately 100–200 μg of each sample was aliquoted and delivered to the Proteomics and Mass Spectrometry Core Facility in the Department of Biochemistry at the University of Colorado, Boulder, for TMT labeling.

Human lumbar spinal cord tissue samples in 8 M urea were reduced and alkylated with the addition of 5% (w/v) sodium dodecyl sulfate (SDS), 10 mM tris(2-carboxyethylphosphine) (TCEP), 40 mM 2-chloroacetamide, 50 mM Tris-HCl, pH 8.5 and incubated shaking at 1000 rpm at room temperature

for 60 min then cleared via centrifugation at 17,000 × g for 10 min at 25 °C. Lysates were digested using the SP3 method (*Hughes et al., 2014*). Briefly, 200 µg carboxylate-functionalized speedbeads (Cytiva Life Sciences) were added to approximately 100 µg protein lysate. Addition of acetonitrile to 80% (v/v) induced binding to the beads, then the beads were washed twice with 80% (v/v) ethanol and twice with 100% acetonitrile. Proteins were digested in 50 mM Tris-HCl buffer, pH 8.5, with 1 µg Lys-C/ Trypsin (Promega) and incubated at 37 °C overnight. Tryptic peptides were desalted using HLB Oasis 1 cc (10 mg) cartridges (Waters) according to the manufacturer's instructions and dried in a speedvac vacuum centrifuge. Approximately 30 µg of the tryptic peptide from each human tissue sample was labeled with TMT 10 plex (Thermo Scientific) reagents according to the manufacturer's instructions. The multiplexed sample was cleaned up with an HLB Oasis 1 cc (10 mg) cartridge. Approximately 50 µg multiplexed peptides were fractionated with high pH reversed-phase C18 UPLC using a 0.5 mm × 200 mm custom packed Uchrom C18 1.8 µm 120 Å (nanolcms) column with mobile phases 10 mM aqueous ammonia, pH10 in water and acetonitrile (ACN). Peptides were gradient eluted at 20 µL/min from 2 to 40% ACN in 40 min concatenating for 12 fractions using a Waters M-class UPLC (Waters). Peptide fractions were then dried in a speedvac vacuum centrifuge and stored at –20 °C until analysis.

## Mass spectrometry analysis

High pH peptide fractions were suspended in 3% (v/v) ACN, 0.1% (v/v) trifluoroacetic acid (TFA) and approximately 1 µg tryptic peptides were directly injected onto a reversed-phase C18 1.7 µm, 130 Å, 75 mm × 250 mm M-class column (Waters), using an Ultimate 3000 nanoUPLC (Thermos Scientific). Peptides were eluted at 300 nL/min with a gradient from 4 to 6% ACN over 120 min then to 25% ACN in 5 min and detected using a Q-Exactive HF-X mass spectrometer (Thermo Scientific). Precursor mass spectra (MS1) were acquired at a resolution of 120,000 from 350 to 1500 m/z with an automatic gain control (AGC) target of 3E6 and a maximum injection time of 50 milliseconds. Precursor peptide ion isolation width for MS2 fragment scans was 0.7 m/z with a 0.2 m/z offset, and the top 15 most intense ions were sequenced. All MS2 spectra were acquired at a resolution of 45,000 with higher energy collision dissociation (HCD) at 32% normalized collision energy. An AGC target of 1E5 and 100 milliseconds maximum injection time was used. Dynamic exclusion was set for 20 s with a mass tolerance of ±10 ppm. Raw files were searched against the Uniprot Human database UP000005640 downloaded November 2, 2020 using MaxQuant v.1.6.14.0. Cysteine carbamidomethylation was considered a fixed modification, while methionine oxidation and protein N-terminal acetylation were searched as variable modifications. All peptide and protein identifications were thresholded at a 1% false discovery rate (FDR).

For visualization of data, likely contaminants, reverse peptides, and proteins quantified by only one peptide were removed. p-values were calculated by Student's t-test (unpaired, homoscedastic variance) combining both non-neurological control samples and combining all ALS cases (including sporadic and *UBQLN2*-mediated).

## Statistical analysis

For western blots, values (normalized to Tubulin and batch-corrected) were compared using the appropriate statistical test by determining normality using a Shapiro-Wilk test and variance using Bartlett's test. For values that were normally distributed and had equal variance, a standard one-way ANOVA was first used to compare differences between the means across all groups. If all groups were not normally distributed a Kruskal-Wallis test was used. For all experimental approaches, appropriate multiple comparison tests were utilized to determine which groups significantly varied. For normal distributions, a Bonferroni's multiple comparisons test was used to compare means directly. For non-normally distributed results, a Dunn's multiple comparisons test was used.

For proteomic analysis, values were compared with an unpaired t-test, and a threshold of $p < 0.05$ was used for significance. For RNA-seq data, statistical analysis is described above using DESeq2 and adjusted p-values.

For all figures, statistical tests are listed in the figure legend and $*p < 0.05$, $**p < 0.01$, $***p < 0.001$, and $****p < 0.0001$.

## Acknowledgements

We would like to acknowledge the Target ALS Human Postmortem Tissue Core, New York Genome Center for Genomics of Neurodegenerative Disease, Amyotrophic Lateral Sclerosis Association, and TOW Foundation for post-mortem tissue samples and RNA-Seq reads. We would also like to acknowledge The Shared Instruments Pool of the Department of Biochemistry at CU Boulder (RRID: SCR_018986), the Biochemistry Cell Culture Core Facility (RRID: SCR_018988), the Biochemistry Flow Cytometry Core Facility (RRID: SCR_019309), and the BioFrontiers Advanced Light Microscopy Core (RRID: SCR_018302) for their assistance with shared equipment. Drs. Miguel Prado and Ramanujan Hegde provided WT, TKO, and *UBQLN2*-rescue HEK293 cells for experiments. We thank Dr. Yoseph Barash and members of his lab for providing us early access to the MAJIQ classifier and for assistance in classifier setup. We thank Dr. Charles Hoeffer for critical discussions of in vivo model design and interpretation of results. We would also like to thank Drs. Edward Chuong, Roy Parker, and Aaron Whiteley for critical reading of our manuscript.

## Additional information

### Competing interests

Alexandra M Whiteley: The University of Colorado, Boulder, has a patent pending for the use of PEG10 inhibitors on which the author is an inventor. The other authors declare that no competing interests exist.

### Funding

| Funder | Grant reference number | Author |
|---|---|---|
| National Institute of General Medical Sciences | T32GM142607 | Julia E Roberts |
| National Cancer Institute | T32CA174648 | G Aaron Holling |
| Biological Sciences Initiative | | Elizabeth Ung |

The funders had no role in study design, data collection and interpretation, or the decision to submit the work for publication.

### Author contributions

Holly H Black, Julia E Roberts, Formal analysis, Validation, Investigation, Methodology, Writing – review and editing; Jessica L Hanson, G Aaron Holling, Autumn M Matthews, Formal analysis, Investigation, Visualization, Methodology, Writing – review and editing; Shannon N Leslie, Conceptualization, Data curation, Formal analysis, Investigation, Methodology, Writing – review and editing; Will Campodonico, Formal analysis, Investigation, Methodology, Writing – review and editing; Christopher C Ebmeier, Conceptualization, Resources, Software, Formal analysis, Methodology, Writing – review and editing; Jian Wei Tay, Resources, Software, Formal analysis, Visualization, Methodology, Writing – review and editing; Elizabeth Ung, Investigation, Methodology; Cristina I Lau, Investigation, Methodology, Writing – review and editing; Alexandra M Whiteley, Conceptualization, Supervision, Funding acquisition, Investigation, Writing – original draft, Project administration, Writing – review and editing

### Author ORCIDs

Holly H Black  http://orcid.org/0000-0002-1383-136X
Jessica L Hanson  http://orcid.org/0000-0001-9165-930X
Will Campodonico  http://orcid.org/0000-0002-9098-5266
Christopher C Ebmeier  http://orcid.org/0000-0001-7940-6190
Jian Wei Tay  http://orcid.org/0000-0002-8634-5039
Cristina I Lau  http://orcid.org/0000-0003-0850-9963
Alexandra M Whiteley  http://orcid.org/0000-0002-4144-7605

### Decision letter and Author response

Decision letter https://doi.org/10.7554/eLife.79452.sa1
Author response https://doi.org/10.7554/eLife.79452.sa2

## Additional files

### Supplementary files
• MDAR checklist

• Source data 1. All raw western blot images.

### Data availability
Figure 6 - Source Data 1 contains the normalized counts from RNA-Seq data used to generate figures. Figure 8 - Source Data 1 contains the abundance counts from proteomics data used to generate figures. Sequencing data have been deposited in the Gene Expression Omnibus (GEO) at GSE227789. Proteomics data is available on PRIDE at PXD031964. Analysis code for microscopy quantitation can be obtained from https://github.com/jwtay1/PEG10-image-analysis/ (copy archived at swh:1:rev:9a772230ad2d6d8913f9619b2b669815c49e76f5). All other data is available in the manuscript or source materials. Correspondence and material requests should be directed to AM Whiteley (alexandra.whiteley@colorado.edu).

The following datasets were generated:

| Author(s) | Year | Dataset title | Dataset URL | Database and Identifier |
|---|---|---|---|---|
| Black HH, Campodonico W, Whiteley AM | 2023 | Gene expression changes in HEK293 cells following PEG10 overexpression | https://www.ncbi.nlm.nih.gov/geo/query/acc.cgi?&acc=GSE227789 | NCBI Gene Expression Omnibus, GSE227789 |
| Leslie SN, Whiteley AM | 2023 | UBQLN2 restrains the domesticated retrotransposon PEG10 to maintain neuronal health in ALS | https://www.ebi.ac.uk/pride/archive/projects/PXD031964 | PRIDE, PXD031964 |

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
