## [Editor Report]

In this manuscript, the investigators provide evidence that levels of Paternally Expressed Gene 10 (PEG10) protein are regulated by Ubqln2 and that proteolytic fragments from PEG10 cleavage induce changes in gene expression, in particular genes that encode proteins involved in axon biology. These data along with their finding that PEG10 levels are increased and alterations of proteins regulated by PEG10 are found in the spinal cord of ALS patients support a role for the abnormal induction of PEG10-regulated genes in ALS.

---

## [Decision Letter]

**Decision letter after peer review:**

Thank you for submitting your article "UBQLN2 restrains the domesticated retrotransposon PEG10 to maintain neuronal health in ALS" for consideration by *eLife*. Your article has been reviewed by 3 peer reviewer, one of whom is a member of our Board of Reviewing Editors, and the evaluation has been overseen by Jeannie Chin as the Senior Editor. The following individual involved in the review of your submission has agreed to reveal their identity: Suzanne Sandmeyer (Reviewer #3).

Essential revisions:

1. It is critical that data directly linking Ubqln2 regulation of PEG10 to ALS pathogenesis be provided. Previous work showed that the Ubqln2-/- mouse model of ALS develops motor deficits. Thus, one means to substantially increase the current work would be if the authors were to show in the Ubqln2-/- mouse that forms of Ubqln2 that alter PEG10- pol levels in cells (Figure 5) improve motor performance in Ubqln2-/- mice. Another approach might be to delete the PPR from PEG10 in a murine model and see if expression in WT mice causes ALS behavior.

2. Figure 1 presents evidence that ubqln2 regulates PEG10 accumulation. The authors use human embryonic stem cells to investigate how knockout (KO) of different ubqln isoforms (1, 2, and 4) affects PEG10 accumulation, showing that only KO of ubqln2 increases the RF1/2 product. However, there is considerable variation in PEG10 expression in the duplicate sample sets provided that is not reflected by the error bars (Figure 1 A and B). For example, RF1/2 is quite different in the two ubqln4 KO lysates, yet the error bars do not capture the variation. Better loading and quantification are needed. Also, in the KO cells, gag levels are slightly increased, which is consistent with alterations in proteasomal degradation. Alternatively, changes in RF1/2 may result from changes in read-through translation, this should be investigated. It would be helpful to include blots showing the lower Mol weight PEG10 products, to see how they change relative to Figure 3.

3. In Figure 1G, the authors examined if removal of the poly proline-rich region (PPR) from PEG10 affects RF1/2 regulation by ubqln, confirming its requirement. Yet, the mechanism why deletion of the PPR abolished RF1/2 regulation by ubqlns was not examined. Is it from accelerated degradation? Also, it is not clear why the authors use the triple ubqln KO cells and did not perform tests in the different ubqln KO cells. The latter comment applies to several of their investigations, leading to uncertainty regarding the specificity of ubqln2 in PEG10 regulation. It is possible that removal of most ubqlns stalls protein degradation affecting PEG10 turnover?

4. To examine the phylogenetic relationship between PEG10 and ubqln2 demonstrating PEG10 levels from marsupials that lack a PPR can be increased by appending a PPR from human PEG10. They used triple ubqln KO cells for these investigations. The change described is not obvious in Figure 2C and E as they appear quite small. They also conclude that ubqln2 regulates PEG10 by these studies, but really the experiments show it is from loss of all ubqlns, not ubqln2 specifically. Thus, the data presented do not support their conclusion in a very convincing manner.

5. The authors show PEG10 is capable of self-cleavage of the RF1 product, generating 2 detectable N-terminal products, and other fragments, including a C-terminal nuclear capsid (NC) fragment (Figure 3). They show expression of HA-tagged NC fragment localizes to mainly the nucleus, whereas several other PEG10 products and fragments localize to the cytoplasm. They provide strong support that PEG10 is capable of self-cleavage by mutation of an aspartate residue (D) in a DSG motif in the protein to alanine (A to → ASG), which abolished cleavage. They also conducted a nice experiment showing the ASG mutant can be cleaved in trans by the introduction of WT PEG10. Unfortunately, the authors never show evidence for liberation and accumulation of the NC fragment, only for an artificially tagged protein by immunofluorescence. The use of a tag to study its localization and effects is problematic as it could influence its properties. They need to show that the fragment is detectable because of their central claim that it is responsible for inducing changes in genes. Biochemical fractionation studies could also reveal the extent of the partitioning of the fragment in the nucleus and cytoplasm. The mechanism by which the NC fragment induces changes in gene expression is not clear.

6. The authors show differences in gene expression upon transfection of HEK293 cells with PEG10 RF1, RF1/2, and NC expression constructs. They show that two PEG10-regulated genes, DCLK1 and TXNIP, are both increased in the spinal cord in sporadic ALS cases compared to controls. It is not clear from these studies whether the changes found in ALS are related to changes in PEG10 specifically, or for other reasons. Additionally, more rigorous comparison in many more ALS and controls is needed. PEG10 levels increase upon cell differentiation (Abed et al.) so the changes in ALS may reflect a compensatory and protective response

7. To investigate if PEG10 RF1/1 levels are altered by ALS mutations in ubqln2 they transfected ubqln TKO cells with either wt ubqln2, or with mutants carrying either the P497H or P506T ALS mutations and found that PEG10 RF1/2 levels are reduced by overexpression of both the wt and P497H mutant, but not by the P506T mutant. They claim that P497H expression did not affect RF1/2 levels. The authors conducted a proteomic comparison of extracts from the spinal cord of two controls, one P497H ubqln2 case, and six sporadic ALS cases, founding increased levels of RF1/2 in the ALS cases. They also found neurofilament medium and neurogranin were both reduced in the ALS cases. Based on these changes they speculate that PEG10 is a novel marker for ALS. The conclusion that the P497S mutant did not affect RF1/2 is incorrect. It reduced RF1/2 slightly more than WT ubqln2. It appears that expression of all three (WT and the 2 ALS mutants) ubqln2 proteins reduces RF1/2 significantly, compared to the TKO cells. In addition, changes in PEG10 found in the ALS cases are difficult to evaluate since very few controls and ALS cases were used for the comparison. Huge variations in the levels of PEG10 and of the other proteins graphed In Fug 6B-F were seen in the two controls. The comparison needs to be done with many more samples for sound statistical comparison. Were the samples compared from the same region of the spinal cord?

8. The authors need to comment on two important points; (1) The cleavage activity of PEG10 has been shown previously – how does this work add to this finding; (2) How do the authors interpret that only one form of fALS affected PEG10?

*Reviewer #1 (Recommendations for the authors):*

1. It is critical that data directly linking Ubiquilin 2 regulation of PEG10 to ALS pathogenesis be provided. Previous work showed that the Ubiquilin 2-/- mouse model of ALS develops motor deficits. Thus, one means to substantially increase the current work would be if the authors were to show in the Ubiquilin 2-/- mouse that forms of Ubiquilin 2 that alter PEG10- pol levels in cells (Figure 5) improve motor performance in Ubiquilin 2-/- mice.

*Reviewer #2 (Recommendations for the authors):*

Additional Experiments

1. Pulse-chase analysis should be conducted to see if any of the changes stem from alterations in protein stability, with and without proteasome inhibitors. It would be nice if the authors examined KD of the different ubqlns in neuronal cells to better relate their findings to ALS. The epitope recognized by the PEG10 antibody could be easily mapped using the deletion constructs in Figure 3.

2. Studies in the absence of only ubqln2, and following the rescue of its expression need to be examined. Effects on changes in protein stability also need to be provided.

3. The mechanism by which the NC fragment induces changes in genes needs clarification.

4. Examination of PEG10 in other human diseases may clarify whether the changes they have observed are specific to ALS.

5. Immunostaining could help clarify and identify the cell types involved in PEG10 induction.

General Comments

1. Many ubqln2 mutations in the PXX domain are known and these need to be examined to reach any conclusion about the domain. Additionally, the PEG10 changes could also be examined in ALS mouse models of ubqln2 and other gene models of ALS.

Overall, more rigorous studies are needed to support the claims made by the authors. As it stands, the current manuscript provides a minor advance in our understanding of PEG10 by ubqlns.

*Reviewer #3 (Recommendations for the authors):*

Because authors demonstrate that PEG10 PPR is required for UBQLN2 activity targeting PEG10, this manuscript would be strengthened if they further implicated UBQLN2 in ALS behavior by showing that PEG10 lacks PPR in mouse model is associated with elevated PEG10 and ALS-like behavioral phenotype.

Of possible interest, the ubiquitin ligase subunit Cdc53 / Hrt1 is a high copy suppressor of Ty3 retrotransposition (Seol et al., 1999). There are some additional similarities to Ty3 that may be of interest to authors. Ty3 Gag3 does not detectably localize to nucleus but similar to PEG10 in this work, Ty3 NC localizes to the nucleus (changes in gene expression not investigated).

Figure 1, Please define the promoter and copy number from which PEG10 is expressed so that it is clear whether these constructs approximate native levels of expression in neuronal tissue. Is there a basis for the choice of HEK (kidney) cell lines for these experiments?

Figure 1. A-C Why is there only one lane of the key protein UBQLN2-/-? If the gel is representative only, then why do B and C have two bar plots for UBQLN1 and 4 rather than one (as does UBQLN2-/-)?

Would make more sense to have the structure diagram of Gag-Pol first rather than fourth place in the figure.

Figure 1 Sup 1 adjust label upward to coincide with migration of protein species UBQLN2,1.

Sup 2 B, PPP > PPR.

Figure 2C, why is the rat/murine PEG10 gag-pol migrating more slowly than other gag-pol species?

P6 l6-8 and Figure 2D,E "PEG10 did not depend on UBQLN expression for its regulation. However, when the human PEG10 PPR was appended to the C-terminus of Koala PEG10, its overall abundance decreased and it became more dependent on UBQLNs for restriction (Figure 2D-E)." This difference which supports the argument that PPR mediates turnover in the Koala example is not very convincing (and is not marked with an asterisk). P6 l8 doesn't really reflect the figure which shows very little difference between koala plus and minus PPR in the presence of UBQLN2

Figure 3C uses modeling for Ty3 Gag from 2007 to propose structure of PEG10 Gag, however, the Ty3 structure was revised based on the ARC protein crystal structure and Ty3 VLP cryoEM in 2019 (Dodonova et al). Authors should revise or remove the suggested structure.

P3, l21 Description of pools of gag and gag-pol may incorrectly create the impression they are separately sequestered and should probably be rephrased.

P6 suggests that authors are exploring PEG10 self-processing for the first time, but this was already demonstrated in Clark et al. (2007) which was cited, but not for the aspartyl protease result. Were experiments regarding cleavage carried out in cells known not to express PEG10? Because the PR can cleave in trans, the native element untagged element could be exhibiting background cleavage of the transfected element.

P7 l20, The statement "unlike traditional retrotransposons, self-cleavage was not a prerequisite for PEG10 VLP formation as gag and gag-Pol asg…" is incorrect. Gag and Gag-Pol (Ty3 and retroviruses) are thought to assemble in order to facilitate PR dimerization and PR activation. Furthermore, multiple AFM and cryoEM studies show the assembly of immature PR- retrovirus and Ty3 mutants. However, maturation reorganization, RT, and IN activities (lacking in PEG10) all do require PR activation.

P11 l3, It would help the reader if authors were more clear about the fALS mutations. For example "Consistent with a loss of function phenotype, mutant UBQLN2P506T-expressing cells failed to restrict PEG10 gag-pol levels (Figure 5A-B), whereas UBQLN2P497H had no effect on gag-pol. " Does this mean that some fALS alleles are associated with increased gag-pol but others are not? Wouldn't this undermine the authors' thesis that fALS UBQLN2 alleles act through loss of PEG10 level control? P11, l11 states that UBQLN2P497 mediated ALS. Other work from Whitely showed that p497 actually has a different phenotype and reduces PEG10? Recheck this

P10 l9 Authors compare changes in gene expression between sALS and the changes in fALS and conclude that there are similar disruptions of gene expression, but do we know that this sALS isolate does not have spontaneous mutations in UBQLN?

Figure 4 should indicate cell type in legend (HEK); Hierarchical heat map looks like duplicates did not cluster? Au comment?

Figure 5 should be a key figure to illustrate increased levels of PEG10 protein in the presence of fALS UBQLN2 but instead, it appears that the dif from wt is minimal despite being starred and that gag-pol is significantly increased in the TKO over the fALS mutant. Are there differences in severity of these UBQLN2 mutant phenotypes that should be described (see above also)? The implication that not all fALS mutations in UBQLN act through PEG10 is counter to the general argument of the authors reflected in the title.

Figure 6 EF, Why is the control a non-neurological sample rather than a non ALS lumbar sample? Au pls clarify how the spike-in PEG10 added to these samples raises the sensitivity without interfering w quantitation of the differences measured. Is "Classical ALS" the same as sALS? IJ The interpretation of the scales and the point of these panels is not clear.

---

## [Author Response]

Essential revisions:1. It is critical that data directly linking Ubqln2 regulation of PEG10 to ALS pathogenesis be provided. Previous work showed that the Ubqln2-/- mouse model of ALS develops motor deficits. Thus, one means to substantially increase the current work would be if the authors were to show in the Ubqln2-/- mouse that forms of Ubqln2 that alter PEG10- pol levels in cells (Figure 5) improve motor performance in Ubqln2-/- mice. Another approach might be to delete the PPR from PEG10 in a murine model and see if expression in WT mice causes ALS behavior.

We appreciate the reviewers’ focus on linking PEG10 to ALS pathogenesis. Unfortunately, it will take >1 year for us to generate data that *Ubqln2*^-/-^ mice can be rescued by PEG10 knockdown or knockout (either of the whole gene or of the PPR region) due to the time it takes for genetic models to develop. Therefore, we considered that in vivo model to be beyond the scope of the current publication.

We have revised our text to reflect the progression of our logic behind the conclusion that PEG10 may be contributing to neurodegenerative disease directly. We have emphasized in our Discussion on Page 21 (lines 7-20) that evidence that PEG10 directly contributes to symptoms of ALS is based on cellular phenotypes observed in HEK cells. We have also put greater emphasis on in vivo data published by Pandya et al. that demonstrate that overexpression of PEG10 leads to changes in neuronal trafficking in the neonatal brain on Page 4, lines 4-7.

Together, we hope that this more thorough discussion of our conclusions adequately addresses this important point.

2. Figure 1 presents evidence that ubqln2 regulates PEG10 accumulation. The authors use human embryonic stem cells to investigate how knockout (KO) of different ubqln isoforms (1, 2, and 4) affects PEG10 accumulation, showing that only KO of ubqln2 increases the RF1/2 product. However, there is considerable variation in PEG10 expression in the duplicate sample sets provided that is not reflected by the error bars (Figure 1 A and B). For example, RF1/2 is quite different in the two ubqln4 KO lysates, yet the error bars do not capture the variation. Better loading and quantification are needed. Also, in the KO cells, gag levels are slightly increased, which is consistent with alterations in proteasomal degradation. Alternatively, changes in RF1/2 may result from changes in read-through translation, this should be investigated. It would be helpful to include blots showing the lower Mol weight PEG10 products, to see how they change relative to Figure 3.

We have broken down the following comment into multiple sections for ease of response.

Apparent variation in hESC blotsApparent increase in gag abundance in hESC blotsPotential changes to read-through translation in different ESC linesVisibility of lower molecular weight PEG10 products in hESC lines by western blot

A. We regret that this Figure appears to have variation and think that much of this is due to a misunderstanding of the set up of this experiment. We have improved our Figure 1A-B with clearer visualization and explanation of the Figure.

We changed the annotation of Figure 1A-C to reflect the fact that there are two independently generated clones of UBQLN1^-/-^ and UBQLN4^-/-^ ESC lines being tested independently, rather than simple duplicates.We changed our bar plot to include all datapoints for transparency.We changed the scale on our gag plot in C to equal that of gag-pol in B, in order to highlight the minimal and non-significant changes to gag upon Ubiquilin perturbation.We have included uncropped images of all western blots used to generate the data in Figure 1A as a Figure Supplement part B to Figure 1.

B. It is mentioned that in KO cells, gag levels are slightly increased. While they appear slightly elevated in *UBQLN1*^-/-^ and *UBQLN4*^-/-^ lines, none of these changes were deemed statistically significant. Therefore, we conclude that we do not see any apparent global alterations in proteasomal degradation. This has been clarified in the text on Page 5, lines 6-8.

C. The best way to determine the rate of readthrough translation in these cells would be to transfect a tagged form of PEG10 into each ES cell line and assess the ratio of gag-pol:gag in the presence of a proteasome inhibitor to eliminate the contribution of gag-pol protein accumulation to the ratio. Because of the difficulty in transfecting ES cells, we instead compared our HEK cell lines (WT versus TKO) for gag-pol:gag ratio and found that even the knockout of UBQLN1, 2, and 4 does not appear to influence the rate of readthrough translation (See Author response image 1). Therefore, we also find it unlikely that there are changes to readthrough translation rate in our single knockout ES cell lines.

**Author response image 1. sa2fig1:** Read-through translation does not appear affected by UBQLN loss, ratio of gag-pol-Dendra2 to gag-Dendra2 to signal in WT or TKO cells treated with bortezomib for 12 hours. Shown is mean ± SEM from three independent experiments. Significance was determined by unpaired Student's T test.

We appreciate the question of whether self-cleavage product bands are visible in hESC western blots. We have included extended images of the original ES cell western blot as Figure Supplement 1B, which do not show any evident cleavage product bands. The PEG10 antibody we use is polyclonal and generated against the entire gag sequence. We have tested this antibody against our PEG10-Dendra2 constructs by western blot and found that it recognizes the smallest cleavage product (see Author response image 2), indicating that this antibody is capable of recognizing an epitope close to the N-terminus of the protein. However, we noticed that it stained these smaller fragments with weak signal intensity despite the overexpression of PEG10 in this test. Therefore, we conclude that the endogenously-formed PEG10 proteolytic fragments are below the limit of detection by western blot in this cell line.

**Author response image 2. sa2fig2:** Proteintech’s polyclonal and PEG10 antibody recognises a very N-terminal portion of PEG10. HEK cells transfected with gag-pol-Dendra2 were transferred by western blot and probed with anti-PEG10 antibody. Even CA^NTD^ can be visualised.

3. In Figure 1G, the authors examined if removal of the poly proline-rich region (PPR) from PEG10 affects RF1/2 regulation by ubqln, confirming its requirement. Yet, the mechanism why deletion of the PPR abolished RF1/2 regulation by ubqlns was not examined. Is it from accelerated degradation? Also, it is not clear why the authors use the triple ubqln KO cells and did not perform tests in the different ubqln KO cells. The latter comment applies to several of their investigations, leading to uncertainty regarding the specificity of ubqln2 in PEG10 regulation. It is possible that removal of most ubqlns stalls protein degradation affecting PEG10 turnover?

We have broken down this comment into multiple sections.

Examination of the reason for PPR’s dependence on UBQLN expressionRationale for the use of TKO cells rather than single knockout

A. We thank the reviewers for the insightful question about how the dPPR’s protein degradation changes compared to full-length gag-pol. The question was asked: Is its degradation accelerated compared to full-length gag-pol protein? In the steady state protein abundance assay, we see that removal of the PPR results in a very small increase in protein abundance in WT cells, and a decrease of protein abundance in TKO cells. This is shown in a new panel of Figure 2H. From this, we conclude that the PPR renders the cell dependent on UBQLNs to regulate the protein’s abundance. However, without the PPR, PEG10 can be degraded by typical proteasomal degradation pathways. A putative model of this difference is shown in Figure 2 Supplement 1B.

We also performed a cycloheximide-based assay on dPPR in WT and TKO cells. This showed that there was still a modest increase in the half-life of PEG10 protein in TKO cells by 8 hours, though there was considerable variability in the behavior of this construct in TKO cells. This is shown in a new panel of Figure 2 Supplement 1C.

Discussion of these new pieces of data is included in the Results section on Page 8-9, lines 14-6.

B. Our rationale for using triple Ubiquilin KO cells is that they naturally express high levels of PEG10 and that the stably integrated, doxycycline-inducible UBQLN2 alleles can be titrated to approximately endogenous levels with restriction of dox in medium. Therefore, these cells serve as an excellent model from which to determine the sufficiency of UBQLN2 to regulate PEG10 degradation. From there, we continued using this model cell line because of the ease of use and establishment as a model system for our purposes.

We have reorganized some of our data to make this rationale more evident. In Figure 1A-C, we start by studying single knockout ES cell lines, which reveal the unique necessity of UBQLN2 to restriction of PEG10 gag-pol. Then, in Figure 1E-G, we show that in HEK cells lacking UBQLN expression, rescue of UBQLN2 is sufficient to reduce PEG10 to WT levels. There, we also introduce information about UBQLN2 mutants and establish the idea that UBQLN2 loss or mutation results in PEG10 gag-pol accumulation (Figure 1F). The Results section has been reorganized accordingly to reflect these changes in organization.

We hope that with these changes, our use of TKO cells as a simple, tractable model for UBQLN essentiality – and its useful service in this case as a proxy for UBQLN2 essentiality – is clearer.

4. To examine the phylogenetic relationship between PEG10 and ubqln2 demonstrating PEG10 levels from marsupials that lack a PPR can be increased by appending a PPR from human PEG10. They used triple ubqln KO cells for these investigations. The change described is not obvious in Figure 2C and E as they appear quite small. They also conclude that ubqln2 regulates PEG10 by these studies, but really the experiments show it is from loss of all ubqlns, not ubqln2 specifically. Thus, the data presented do not support their conclusion in a very convincing manner.

We regret that this data was presented in a manner that was not very convincing. We have changed the presentation of this data and have changed text to reflect the nuanced nature of our results.

Visually, we have changed the visualization of our data to break down our interpretation of the data into discrete steps. In Figure 3D-E, we show biological replicates of four independent experiments, rather than the amassed technical and biological replicates that we had originally shown, to highlight the Dendra/CFP signal for each species. We also split the visualization of these results between placental mammals and marsupials, which are now parts D and E, respectively. The use of biological replicates slightly changed the statistical analysis to where every placental mammal shows a significant elevation of protein abundance in TKO cells, whereas marsupials show no significant differences, even with the addition of the human PPR sequence at the C-terminus (Figure 3E). However, we find two aspects of this data notable. First, that addition of the PPR to Koala PEG10 resulted in a dramatic decrease in its abundance, meaning that the PPR introduces a means of protein regulation independent of UBQLN2 in this situation. Second, that even though the Dendra2/CFP ratios are not significantly different between WT and TKO cells +PPR in Figure 3E, the ratio of TKO/WT values is consistently above 1 +PPR, whereas standard Koala PEG10 has a ratio just below 1. This difference is significant as measured by Student’s paired T test, which is now highlighted in Figure 3F.­­­

On Pages 9-10 lines 20-3, we have changed the emphasis of this data to instead reflect the complementary nature of the phylogenetic results to our data in Figure 2H-I showing that the PPR confers some UBQLN dependency to PEG10. We have emphasized that adding a PPR to Koala PEG10 decreases its abundance in WT cells, which is consistent with our data showing that removal of a PPR in human PEG10 increases its abundance in WT cells. We have also emphasized that the ratio of TKO/WT abundance is significantly different for Koala PEG10 lacking and including the human PPR region, consistent with a partial role for the PPR in contributing to UBQLN-dependent PEG10 degradation. Finally, we have removed the statement speculating on the evolutionary relationship between UBQLN2 and PEG10 given the subtle nature of our phylogenetic data.

5. The authors show PEG10 is capable of self-cleavage of the RF1 product, generating 2 detectable N-terminal products, and other fragments, including a C-terminal nuclear capsid (NC) fragment (Figure 3). They show expression of HA-tagged NC fragment localizes to mainly the nucleus, whereas several other PEG10 products and fragments localize to the cytoplasm. They provide strong support that PEG10 is capable of self-cleavage by mutation of an aspartate residue (D) in a DSG motif in the protein to alanine (A to → ASG), which abolished cleavage. They also conducted a nice experiment showing the ASG mutant can be cleaved in trans by the introduction of WT PEG10. Unfortunately, the authors never show evidence for liberation and accumulation of the NC fragment, only for an artificially tagged protein by immunofluorescence. The use of a tag to study its localization and effects is problematic as it could influence its properties. They need to show that the fragment is detectable because of their central claim that it is responsible for inducing changes in genes. Biochemical fractionation studies could also reveal the extent of the partitioning of the fragment in the nucleus and cytoplasm. The mechanism by which the NC fragment induces changes in gene expression is not clear.

We have broken down this comment into multiple sections for ease of response.

Evidence for liberation and accumulation of NC fragment from cleavage-competent PEG10Exploration of the mechanism by which the NC fragment induces gene expression changes

A. We agree that tags can sometimes compromise the activity of a given protein, but we do not think this is the case for HA-NC because the identical HA tag was appended to all forms of PEG10 and did not universally induce nuclear localization for gag-pol or gag, indicating that something unique about the NC sequence drives its localization.

That being said, we completely agree on the value of detecting endogenously cleaved NC fragment in cells. We have visualized the generation of tag-less NC fragment from gag-pol, but not gag-pol^ASG^, using a custom-made NC antibody. The antibody was generated against gag AA260-325; therefore, it will recognize NC fragment as well as any PEG10 protein (gag, gag-pol) containing the NC region. We first visualized the cleaved NC fragment by western blot of whole cell lysate, which showed that a NC-staining, 10 kDa fragment is only present in cells expressing PEG10 gag-pol, and not gag-pol^ASG^, indicating that the presence of this fragment is dependent on an intact protease domain. We take this as evidence for self-cleavage of PEG10 gag-pol to generate an NC fragment. This data is shown as a new panel in Figure 4I.

To determine the contribution of the HA-tag to HA-NC localization, we removed the HA tag and performed the same immunofluorescence using our custom NC antibody. We saw an identical staining pattern, indicating that the HA tag is not influencing localization of this protein fragment. This data is included as a new panel of Figure 5B.

B. The mechanism by which the NC fragment induces gene expression changes remains unclear. We have performed some experiments that explored two favorable hypotheses: changes to mRNA splicing and changes to mRNA trafficking. However, we found no significant differences in either phenomenon upon PEG10 transfection. We consider further examination of this particular aspect of PEG10 function to be beyond the scope of the current study. We have added to the text on Page 14 lines 15-23 that briefly summarize our negative data. Further speculation of NC’s mechanism of action is discussed on Page 20 lines 14-18.

6. The authors show differences in gene expression upon transfection of HEK293 cells with PEG10 RF1, RF1/2, and NC expression constructs. They show that two PEG10-regulated genes, DCLK1 and TXNIP, are both increased in the spinal cord in sporadic ALS cases compared to controls. It is not clear from these studies whether the changes found in ALS are related to changes in PEG10 specifically, or for other reasons. Additionally, more rigorous comparison in many more ALS and controls is needed. PEG10 levels increase upon cell differentiation (Abed et al.) so the changes in ALS may reflect a compensatory and protective response

We have broken down this comment into multiple sections for ease of response.

Clarification of source of changes to gene expression in sALS tissue samplesImprovement of ALS data with more ALS cases and controls.

A. We agree with the reviewers that it is impossible to determine from these patient samples whether the similar increases observed in *DCLK1* and *TXNIP* are a result of PEG10 specifically, or due to other reasons (such as a compensatory/protective response, as suggested). We included this data from patient samples to highlight the shared transcriptional changes between these two very different populations – transfected cells and patient tissues – but agree that we should fully acknowledge that the source of these transcriptional changes in patient tissues is unresolved. We have changed the text of our Results on Page 14 lines 8-11 to reflect that we are examining alteration of transcript abundance in sALS tissue, rather than whether these transcript changes contribute to disease. We have also emphasized the future work necessary to tease apart these changes in our text on Page 21 lines 7-17.

B. We appreciate the reviewer’s comment that many more ALS cases and controls were needed. We reached out to Target ALS to improve our RNA-Seq dataset and were able to increase the number of datapoints included. Our updated RNA-Seq tissue sample cohort is the most comprehensive dataset of RNA-Seq samples available in the US as of Fall 2022 and includes 17 ‘non-neurological controls’ (post-mortem samples where the cause of death was something other than a neurological disease), 127 ALS cases, and 2 known *UBQLN2*-mediated cases. We performed an identical RNA-Seq analysis on this updated cohort and received similar results for *DCLK1*, *TXNIP*, and *PEG10* transcript abundance, thereby increasing the significance of the difference between ALS and controls. This data has now replaced the original data in Figure 6G and Figure 7G. Our Methods section has also been updated on Page 37 lines 6-11 to reflect these updates.

7. To investigate if PEG10 RF1/1 levels are altered by ALS mutations in ubqln2 they transfected ubqln TKO cells with either wt ubqln2, or with mutants carrying either the P497H or P506T ALS mutations and found that PEG10 RF1/2 levels are reduced by overexpression of both the wt and P497H mutant, but not by the P506T mutant. They claim that P497H expression did not affect RF1/2 levels. The authors conducted a proteomic comparison of extracts from the spinal cord of two controls, one P497H ubqln2 case, and six sporadic ALS cases, founding increased levels of RF1/2 in the ALS cases. They also found neurofilament medium and neurogranin were both reduced in the ALS cases. Based on these changes they speculate that PEG10 is a novel marker for ALS. The conclusion that the P497S mutant did not affect RF1/2 is incorrect. It reduced RF1/2 slightly more than WT ubqln2. It appears that expression of all three (WT and the 2 ALS mutants) ubqln2 proteins reduces RF1/2 significantly, compared to the TKO cells. In addition, changes in PEG10 found in the ALS cases are difficult to evaluate since very few controls and ALS cases were used for the comparison. Huge variations in the levels of PEG10 and of the other proteins graphed In Fug 6B-F were seen in the two controls. The comparison needs to be done with many more samples for sound statistical comparison. Were the samples compared from the same region of the spinal cord?

We have broken down this comment into multiple sections for ease of response.

Discussion of UBQLN2 mutation data in cells versus in tissue samplesLimitation of conclusions based on number of samples in proteomic datasetClarification of the region of spinal cord selected for study

A. We regret the confusion regarding the effects of the P497H mutation on PEG10 levels. With regards to the statement that P497S decreases RF1/2 compared to WT cells, this was not significant by our metrics. According to multiple comparisons T test performed on data in Figure 1F, the difference in gag-pol abundance between WT UBQLN2 and P497H expression is insignificant. This has now been annotated in the Figure for clarity.

Based on data presented in Figure 1F, we agree with the statement that ‘it appears that expression of all three ubqln2 proteins reduces RF1/2 significantly compared to the TKO cells’. We would conclude that the UBQLN2 mutations present in disease confer a partial loss of function with regards to gag-pol degradation. We have included text reflecting this point on Page 6, lines 1-3.

B. We agree that as a rule, it is always ideal to include as many samples as possible in a proteomic dataset. This dataset consists of samples obtained from post-mortem lumbosacral spinal cord from either non-neurological controls (NNC, named as such because ‘healthy’ is an inappropriate word to describe post-mortem samples; however it was confirmed that these individuals did not have any diagnosed neurological condition) or ALS cases. After considerable work with nonprofit foundations, we were only able to secure two matched NNC samples’ unfixed lumbosacral spinal cords to go with our ALS samples. Overall, the legal process to obtain these samples took almost one year. Therefore, we are unable to add to this proteomic dataset in any reasonable amount of time.

However, we would like to emphasize the uniqueness of this dataset in the literature. Human spinal cord proteomic datasets are rare, and an n = 7 is quite substantial. Even rodent proteomics have used an n = 5 per group^1^, or n = 3 of pooled samples^2^. Of course, more samples would be ideal, but it is notable that we were able to amass the samples that we did.

Nelvagal et al., “Comparative Proteomic Profiling Reveals Mechanisms for Early Spinal Cord Vulnerability in CLN1 Disease.” *Scientific Reports* (2020).Yang et al., “Proteomic Analysis of Spinal Cord Tissue in a Rat Model of Cancer-Induced Bone Pain.” *Frontiers in Molecular Neuroscience* (2022).

Fixed histological sections are easier to obtain, but do not provide the same opportunity for detailed examination of gag versus gag-pol protein quantitation, because no antibody exists that reliably detects the pol region of PEG10, and the cross-linking of proteins caused by fixation precludes us from performing proteomic analysis of these samples. When we initially performed immunofluorescence staining of PEG10 in fixed tissue sections (now in Figure 7A-C), we saw detectable PEG10 signal in both non-neurological control samples and ALS samples that was very similar when quantified. This data is included in Figure 7. But because this antibody detects both gag and gag-pol, we were unable to discern unique changes to gag-pol protein in disease. Western blots of spinal cord tissue are not clean enough to be able to see PEG10, so we instead performed the work-intensive global proteomics to pull out PEG10 gag-pol abundance. This is discussed on Page 15 lines 13-18.

C. We regret not making this clear in our manuscript: the spinal cord sections used for proteomics and histological staining were all collected from the lumbosacral region. We have updated the Methods on Page 27 lines 12-14 and main text on Page 15 line 20 to reflect this emphasis.

8. The authors need to comment on two important points; (1) The cleavage activity of PEG10 has been shown previously – how does this work add to this finding; (2) How do the authors interpret that only one form of fALS affected PEG10?

(1) We appreciate the reviewers request for us to clarify how our work adds to previous findings of PEG10 cleavage activity. The main study that touches on PEG10 self-cleavage is Clark et al. 2007. It is also mentioned and likely observed by Lux et al. 2005 (though not directly tested with a protease-dead mutant), as well as Golda et al. 2020. These previous findings are highlighted in Page 10 lines 12-14 of our Results section and Pages 19-20, lines 22-1 of our Discussion. Our study improves on these findings in three ways. First, we used a tagged form of PEG10 to observe self-cleavage with much greater clarity than the previous manuscripts. Second, we were able to determine the likely locations of PEG10 self-cleavage events through a combination of biochemical methods. By determining the precise cleavage sites, we were uniquely able to explore the consequences of fragment generation on protein fate. Finally, our study is the first to examine the functional consequences of self-cleavage with the generation of NC and its effects on the cell. These points are mentioned on Page 20, lines 2-6 of our Discussion.

(2) Concerning the second point, we should clarify that while P497H did not mimic P506T in the accumulation of endogenous PEG10 compared to WT rescue in HEK cells, this may not reflect what happens in human disease. In fact, as pointed out by reviewers above, we observed elevated PEG10 pol levels in the spinal cord of P497H-associated fALS. Furthermore, we found that P497H, as well as P506T, were significantly less able to restrain overexpressed PEG10-Dendra2 levels in TKO HEK cells compared to WT counterparts, indicating that P497H does share some defective activity with P506T. Discussion of this new data, shown in Figure 2D-E, are mentioned in the Results section on Pages 7 lines 9-14. Discussion of the unique properties of each allele are on Page 18 lines 4-13 of the Discussion.